# Skirting Additive Error Barriers for Private Turnstile Streams

**Anders Aamand**
BARC
University of Copenhagen
Copenhagen, Denmark
aa@di.ku.dk

**Justin Y. Chen**
CSAIL, EECS
Massachusetts Institute of Technology
Cambridge, MA, USA
justc@mit.edu

**Sandeep Silwal**
Department of Computer Sciences
UW Madison
Madison, WI, USA
silwal@cs.wisc.edu

## Abstract

We study differentially private continual release of the number of distinct items in a turnstile stream, where items may be both inserted and deleted. A recent work of Jain, Kalemaj, Raskhodnikova, Sivakumar, and Smith (NeurIPS '23) shows that for streams of length $T$, polynomial additive error of $\Omega(T^{1/4})$ is necessary, even without any space restrictions. We show that this additive error lower bound can be circumvented if the algorithm is allowed to output estimates with both additive *and multiplicative* error. We give an algorithm for the continual release of the number of distinct elements with $\mathrm{polylog}(T)$ multiplicative and $\mathrm{polylog}(T)$ additive error. We also show a qualitatively similar phenomenon for estimating the $F_2$ moment of a turnstile stream, where we can obtain $1 + o(1)$ multiplicative and $\mathrm{polylog}(T)$ additive error. Both results can be achieved using polylogarithmic space whereas prior approaches use polynomial space. In the sublinear space regime, some multiplicative error is necessary even if privacy is not a consideration. We raise several open questions aimed at better understanding trade-offs between multiplicative and additive error in private continual release.

## 1 Introduction

Differential privacy (DP) under continual release captures the setting where private data is updated over time. The data arrives one at a time in a stream, and an algorithm must privately release an underlying statistic of interest as the data changes. The continual release model goes back to the early days of DP and has been extensively studied for counting the number of events in a binary stream, also called continual counting Dwork et al. (2010); Chan et al. (2011); Honaker (2015); Andersson & Pagh (2023); Fichtenberger et al. (2023); Henzinger et al. (2023); Andersson et al. (2024); Henzinger et al. (2024b); Dvijotham et al. (2024); Henzinger & Upadhyay (2025), private machine learning applications (via DP-SGD and DP-FTRL) Kairouz et al. (2021); Denissov et al. (2022); McMahan & Thakurta (2022); Choquette-Choo et al. (2023a;b; 2024) (also see the survey of Pillutla et al. (2025)), and tracking graph statistics Song et al. (2018); Fichtenberger et al. (2021); Jain et al. (2024); Raskhodnikova & Steiner (2024); Aryanfard et al. (2025); Andersson et al. (2026). In our setting, the stream is modeled as items from a known universe $[n]$ of $T$ updates. We consider the *turnstile* model where items may be both inserted and deleted.

Continual release is algorithmically interesting since it combines challenges of both differential privacy and streaming algorithms, and there is a recent line of work aiming to nail down the optimal privacy-utility trade-offs for continual release of some of the most basic and well-studied statistics in a stream, including estimating the number of distinct elements and the $F_p$ moments of the stream Epasto et al. (2023); Jain et al. (2023b;a); Henzinger et al. (2024a); Cummings et al. (2025);

Aryanfard et al. (2025); Andersson et al. (2026); Epasto et al. (2026). We focus on the distinct elements and $F_2$ moment estimation problems in this paper, two of the foundational problems in streaming algorithms.

However, there remain substantial gaps in our understanding of these fundamental streaming problems when privacy is a concern. Even ignoring space considerations, which make the aforementioned problems trivial in the standard streaming setting without privacy, there exist polynomial gaps (in the stream length $T$) between known upper and lower bounds in turnstile streams. For example, for estimating the number of distinct elements[1] in the stream, the best known algorithms achieve a $\tilde{O}(T^{1/3})$[2] additive error bound Jain et al. (2023a); Cummings et al. (2025). On the flip side, it is known that any private algorithm *must* incur $\Omega(T^{1/4})$ additive error Jain et al. (2023a), and closing this gap is a challenging open problem. Furthermore, just from sensitivity considerations, it is easy to see that any algorithm for private $F_2$ estimation[3] must incur $\Omega(T)$ additive error.

We are motivated by the fact that (low-space) streaming algorithms for both distinct elements and $F_2$ estimation must incur *multiplicative* error Jayram & Woodruff (2013), and thus it is natural to ask if one can go beyond the existing additive error lower bounds for continual release in turnstile streams if the algorithm is allowed to output estimates with both *multiplicative and additive* error. Indeed, some evidence of why this is possible is already present in the prior work of Epasto et al. (2023) which obtains polylogarithmic additive error along with small multiplicative error for $F_p$ moment estimation, including distinct elements, albeit in the significantly easier setting of insertion-only streams where items are never deleted. Furthermore, it can be checked that the lower bound instances for $\Omega(T^{1/4})$ additive error for distinct elements Jain et al. (2023a) and $\Omega(T)$ additive error for $F_2$ estimation both occur when the true underlying value is itself much larger than the additive error, meaning they do not imply any hardness for obtaining constant multiplicative error. Beyond the continual release setting, overcoming purely additive error lower bounds by introducing multiplicative error has been a feature of several recent works in DP Aamand et al. (2025); Ghazi et al. (2025); Aryanfard et al. (2025).

The main conceptual message of our paper is that polynomial additive errors for fundamental streaming problems can be replaced with **polylogarithmic additive errors, at the cost of some multiplicative error**. Furthermore this can often be achieved while simultaneously using *small space*.

## 1.1 OUR RESULTS

We are focused on computation over data streams of length $T$ from a universe of size $n$. We use $(a_1, s_1), \ldots, (a_T, s_T)$ to denote a *general turnstile* data stream where $a_i \in [n]$ is an element identifier and $s_i \in \{-1, 0, 1\}$ is the increment amount. We call an update an insertion if $s_i = 1$ and a deletion if $s_i = -1$. In *insertion-only* streams, $s_i = 1$ and in *strict turnstile* streams, at any point in time, the number of deletions to any given element can never exceed the insertions to that element (the element's frequency cannot be negative). We are concerned with the following notion of mixed multiplicative and additive error for continual estimation of stream statistics.

**Definition 1.1** (Multiplicative and Additive Error for Continual Estimation). *Let $Y_t \in \mathbb{R}$ be a function of the prefix of a stream $(a_1, s_1), \ldots, (a_t, s_t)$. A streaming algorithm for the continual estimation problem outputs an estimate $\hat{Y}_t$ after receiving the $t$th stream update for all $t \in [T]$. For parameters $\alpha \geq 1, \beta \geq 0$, we say the algorithm solves the problem with $(\alpha, \beta)$ error if there exist parameters $p, q \geq 1$ with $pq = \alpha$ and $r, s \geq 0$ with $r + s = \beta$ such that, for all $t \in [T]$,*

$$Y_t/p - r \leq \hat{Y}_t \leq qY_t + s.$$

*For a randomized algorithm, we say the algorithm has error $(\alpha, \beta)$ with probability $1 - \gamma$ if the error bounds hold across all timesteps with probability at least $1 - \gamma$ over the randomness of the algorithm.*

If an algorithm satisfies $(1, \beta)$ error, we say it has purely additive error. If an algorithm satisfies $(\alpha, 0)$ error, we say it has purely multiplicative error.

---

[1] The number of elements with non-zero frequency.

[2] We use $\tilde{O}(f)$ to denote $O(f \cdot \mathrm{polylog}(f))$.

[3] The $F_2$ value of a stream, also known as the second frequency moment, is defined as the sum of squares of frequencies of items in the stream.

**Remark 1.1** (Interpretation of $(\alpha, \beta)$ error). *One interpretation of $(\alpha, \beta)$ error is that we get close to an $\alpha$ multiplicative approximation to the statistic $Y_t$ as long as it is above the noise floor $Y_t \gg \beta$.*

Our results in this paper operate in the *event-level* notion of privacy for continual release. In this setting, two datasets are neighboring if they differ only in a single update $(a_i, s_i)$. This contrasts with the more stringent notion of privacy called *item-level* in which two datasets differ in all updates which touch a certain domain element $a \in [n]$. The formal definition of privacy can be found in Section 2.

**Distinct Elements** When the statistic of interest is the number of distinct (non-zero frequency) elements, we show that we can skirt the lower bound of $\Omega(T^{1/4})$ from Jain et al. (2023a) if we also allow multiplicative error in our estimates. Our main result is an algorithm that solves the distinct elements problem with only polylogarithmic additive and multiplicative error.

**Theorem 1.1** (Informal version of Theorems 3.1 and 4.1). *Let $\varepsilon, \delta < 1$. There exists an $(\varepsilon, \delta)$-DP algorithm for the continual distinct elements problem that with probability $1 - 1/poly(T)$, for all points in time $t \in [T]$ outputs an estimate $\tilde{D}_t$ of the number of distinct elements $D_t$ with error $(\alpha, \beta)$ where $\alpha, \beta = O\left(\frac{polylog(T, 1/\delta)}{\varepsilon}\right)$. The space usage of the algorithm is $polylog(n, T)$.*

Recall that the best bound from prior work incurs error $(1, \tilde{O}(T^{1/3}))$, ignoring dependencies on $\varepsilon$ and $\delta$. Our bound improves upon this as long as the true number of distinct elements is upper bounded by $\tilde{O}(T^{1/3})$. As $n$ is a trivial upper bound for the additive error of any algorithm (the number of distinct elements is between $0$ and $n$), the prior work gives non-trivial estimates when the stream length is upper bounded as $T \ll n^3$. By comparison, our bounds give non-trivial estimates for $T$ which grows superpolynomially in $n$. In settings where our bound and the prior bound alternate in quality over the course of the stream, the best of the two bounds (up to constant factors) can be achieved by simply running both in parallel each with half of the privacy budget and outputting the estimate from prior work when it exceeds twice its additive error.

Our results are achieved by two different algorithms for continual distinct elements estimation. As a core primitive, both rely on differentially private continual counting. Our contribution is using tools from the streaming literature to carefully transform distinct elements estimation into certain counting problems which are amenable to the additive error guarantees of private continual counting.

The first algorithm (which leads to Theorem 3.1) is inspired by the classic idea of using the minimum hash value of keys in a set $A$ to estimate the size of $A$ and appears in Section 3. Under privacy constraints, we cannot compute the minimum hash value exactly. Instead, we create buckets based on the least significant non-zero bit of hashes of keys such that the expected number of keys hashing to the buckets increase geometrically. We can then use private continual counting in each bucket to approximately determine the min hash.

The second algorithm (which yields Theorem 4.1) is based on performing a domain reduction, also via a hash function, to a domain sufficiently small such that many elements collide. These collisions can be detected by a private continual counting algorithm. We can then use the size of the reduced domain as an estimate for the number of distinct elements. This algorithm is presented in Section 4.

The minimum hash algorithm is limited to strict turnstile streams but achieves better error and less space usage than the domain reduction algorithm which applies to general turnstile streams. In Section 4, we also show a potential path towards achieving arbitrarily good multiplicative error. Using the same tools as the domain reduction algorithm, we show the following reduction: a (hypothetical) algorithm achieving purely additive error sublinear in the domain size $n$ implies the existence of an algorithm achieving $(1 + \eta, polylog(T))$ error. See Theorem 4.2 for details.

**Frequency Moments** In addition to the distinct elements problem, we also consider the $F_2$ estimation problem, where we are interested in approximating the second moment of the frequencies of elements in the stream at any point of time. Specifically, if each update $(a_t, s_t)$ satisfies $a_t \in [n]$ and $s_t \in \{-1, 0, 1\}$, we may define $x_t[j] = \sum_{i \leq t : a_i = j} s_j$, and we are interested in approximating $F_2 = \sum_{j \in [n]} x_t[j]^2$ for any $t$. It is easy to see that any algorithm with only purely additive error $(\alpha, \beta) = (1, \beta)$, must have $\beta = \Omega(T)$, simply because the sensitivity of the second moment is $\Omega(T)$. Surprisingly, we show that allowing a small constant multiplicative error, the additive error

| Source | Error | Space | Privacy | Notes |
|---|---|---|---|---|
| Jain et al. (2023b) | $(1, \tilde{O}(T^{1/3}))$ | $O(T)$ | Item-level | —— |
| Cummings et al. (2025) | $(1 + \eta, \tilde{O}(T^{1/3}))$ | $\tilde{O}_\eta(T^{1/3})$ | Event-level | —— |
| Jain et al. (2023a) | $(1, \tilde{\Omega}(T^{1/4}))$ | —— | Event-level | Lower bound |
| Epasto et al. (2023) | $(1 + \eta, O_\eta(\log^2(T)))$ | polylog$(T)$ | Event-level | Insertion-only |
| Theorem 3.1 | $(O(\log^2(T)), O(\log^2(T)))$ | $O(\log^3(T))$ | Event-level | Strict turnstile |
| Theorem 4.1 | $(O(\log^{10}(T)), O(\log^{10}(T)))$ | poly$(T)$ | Event-level | —— |

Table 1: Multiplicative and additive error bounds for $(\varepsilon, \delta)$-DP algorithms for the Continual Distinct Elements Problem. We ignore dependencies on the privacy parameters. Unless otherwise stated, upper bounds hold for general turnstile streams and lower bounds hold for strict turnstile streams. We present results which hold for worst-case streams of length $T$. See Appendix A for numerous prior works which give instance-specific bounds depending on stream statistics.

| Source | Error | Space | Privacy | Notes |
|---|---|---|---|---|
| Epasto et al. (2023) | $(1 + \eta, \tilde{O}_\eta(\log^7(T)))$ | $O_\eta(\log^2(T))$ | Event-level | Insertion-only |
| Lemma 5.1 | $(1, \Omega(T))$ | —— | Event-level | Lower bound |
| Theorem 5.1 | $(1 + \eta, \tilde{O}_\eta(\log^4(T)))$ | $O_\eta(\log^2(T))$ | Event-level | —— |

Table 2: Multiplicative and additive error bounds for $(\varepsilon, \delta)$-DP algorithms for the Continual $F_2$ Problem. We ignore dependencies on the privacy parameters. Unless otherwise stated, upper bounds hold for general turnstile streams and lower bounds hold for strict turnstile streams.

can be made polylogarithmic. Furthermore, our algorithm quantitatively improves upon the prior work of Epasto et al. (2023) which only applies in the insertion-only model.

**Theorem 1.2** (Informal version of Theorem 5.1)**.** *Let $\varepsilon, \delta, \eta < 1$. There exists an $(\varepsilon, \delta)$-DP algorithm for the $F_2$ estimation problem that with probability $1 - 1/poly(T)$, for all points in time $t \in [T]$ outputs an estimate of $\tilde{F}_2$ of $F_2$ with error $(1 + \eta, \beta)$ where $\beta = polylog(T, \eta, \delta)/(\varepsilon^2 \eta^3)$. The space usage of the algorithm is $polylog(T)/\eta^2$.*

Again, this result relies on continual counting. We use the Johnson-Lindenstrauss reduction to map the $n$-dimensional frequency vector to a small domain, and use continual counting to estimate the coordinates in the reduced domain.

### 1.2 PRIOR WORK AND OPEN PROBLEMS

We include detailed discussion of prior and concurrent works in Appendix A and several interesting open questions in Appendix B.

## 2 PRELIMINARIES AND NOTATION

Throughout the paper, poly$(T)$ refers to a polynomial of arbitrarily large constant degree. We consider the base 2 logarithm by default: $\log(\cdot) = \log_2(\cdot)$.

### 2.1 DATA STREAMS

Let $(a_1, s_1), \ldots, (a_T, s_T)$ denote the data stream where $a_i \in [n]$ is an element identifier and $s_i \in \{-1, 0, 1\}$ is the increment amount. Let $x_t \in \mathbb{R}^n$ denote the frequency vector at time step $t$. For $i \in [n]$, $x_t[i]$ is the sum of increments to item $i$ across all timesteps up to $t$. In the *strict turnstile* model, we have $x_t[i] \geq 0$ for all $i \in [n], t \in [T]$, and the number of distinct elements at time $t$ is defined as $D_t = |i \in [n] : x_t[i] > 0|$. In the *general turnstile* model, the true frequency vector $x_t$ is allowed to have negative entries and the number of distinct elements is simply the number of non-zero coordinates. We interchangeably denote the number of distinct elements as $\|x_t\|_0$.

In this work, we assume that the universe size $n$ and stream length $T$ are known up to constant factors to the streaming algorithm a priori for simplicity. Without loss of generality, we will consider $n = \text{poly}(T)$ by standard hashing tricks.

A streaming algorithm processes each update one at a time while maintaining only a bounded memory. In this work, we measure space in terms of words of size at least $\Omega(\log T)$ bits. This is a standard measurement, assuming that the length of the stream and the universe size can be stored in a constant number of words. We also assume access to an oracle which can produce a sample of one-dimensional Gaussian noise $\mathcal{N}(0, \sigma^2)$ which can be stored in constant words. See Canonne et al. (2022) for background on sampling Gaussian noise for differential privacy.

We consider randomized streaming algorithms with non-adaptive adversaries. The stream is chosen independent of the randomness of the algorithm. While some private algorithms can succeed against adaptive adversaries Jain et al. (2023b), the best (even non-private) streaming algorithms for many fundamental problems including distinct elements require space polynomial in the stream length Attias et al. (2024).

## 2.2 DIFFERENTIAL PRIVACY

**Definition 2.1** (Event-Level Neighboring Datasets). *Let $X = (a_1, s_1), \ldots, (a_T, s_T)$ and $X' = (a'_1, s'_1), \ldots (a'_T, s'_T)$ be two strict turnstile data streams. These streams are neighboring if there exists an index $i \in [T]$ such that $(a_i, s_i) \neq (a'_i, s'_i)$, and for all $j \neq i$, $(a_j, s_j) = (a'_j, s'_j)$.*

A different notion of item-level privacy is also studied where all updates which touch a given element $a \in [n]$ may change between neighboring dataset. In this work, we focus on the event-level definition.

**Definition 2.2** (Differential Privacy Dwork et al. (2006)). *Let $\mathcal{A} : \mathcal{X} \to \mathcal{Y}$ be a randomized algorithm. For parameters $\varepsilon, \delta \geq 0$, $\mathcal{A}$ satisfies $(\varepsilon, \delta)$-DP if, for any two neighboring datasets $X, X' \in \mathcal{X}$ and measurable subset of outputs $O \subseteq \mathcal{Y}$,*

$$\mathbf{Pr}[\mathcal{A}(X) \in O] \leq e^\varepsilon \mathbf{Pr}[\mathcal{A}(X') \in O] + \delta.$$

We will also make use of a similar form of differential privacy called zero-concentrated differential privacy, $\rho$-zCDP, introduced by Bun & Steinke (2016). $\rho$-zCDP implies $(\varepsilon, \delta)$-DP and benefits from tighter and simpler composition of multiple private mechanisms.

**Definition 2.3** (Rényi Divergence Rényi (1961)). *Given two distributions $P, Q$ over $\mathcal{Y}$ and a parameter $\zeta \in (1, \infty)$, we define the Rényi Divergence as*

$$D_\zeta(P\|Q) = \frac{1}{\zeta - 1} \log\left( \mathbf{E}_{z \sim P}\left[ \left( \frac{P(z)}{Q(z)} \right)^{\zeta - 1} \right] \right).$$

**Definition 2.4** (Zero-Concentrated Differential Privacy (zCDP) Bun & Steinke (2016)). *Let $\mathcal{A} : \mathcal{X} \to \mathcal{Y}$ be a randomized algorithm. For parameter $\rho \geq 0$, $\mathcal{A}$ satisfies $\rho$-zCDP if, for any two neighboring datasets $X, X' \in \mathcal{X}$ and $\zeta \in (1, \infty)$,*

$$D_\zeta(\mathcal{A}(X)\|\mathcal{A}(X')) \leq \rho\zeta.$$

**Lemma 2.1** (zCDP Composition Bun & Steinke (2016)). *Let $\mathcal{A} : \mathcal{X} \to \mathcal{Y}$ and $\mathcal{B} : \mathcal{X} \times \mathcal{Y} \to \mathcal{Z}$ be algorithms satisfying $\rho_1$-zCDP and $\rho_2$-zCDP, respectively. Then, the algorithm which, given an input $X \in \mathcal{X}$, outputs $(\mathcal{A}(X), \mathcal{B}(X, \mathcal{A}(X)))$ satisfies $(\rho_1 + \rho_2)$-zCDP.*

**Lemma 2.2** (zCDP Translation Bun & Steinke (2016)). *Any algorithm satisfying $\rho$-zCDP also satisfies $(\rho + 2\sqrt{\rho \log(1/\delta)}, \delta)$-DP for any $\delta > 0$.*

Therefore, an algorithm satisfying $\rho$-zCDP also satisfies $(\varepsilon, \delta)$-DP if $\rho = O(\varepsilon^2 / \log(1/\delta))$.

The simpler problem of DP Continual Counting was introduced in Dwork et al. (2010); Chan et al. (2011). In this setting, the stream is comprised of a sequence of updates in $\{-1, 0, 1\}$ to an underlying count and the goal is to maintain a running approximation to the prefix sum of updates at every timestep with small additive error. The problem is normally defined over a stream of bits with $b_i \in \{0, 1\}$. Allowing decrements as well can be achieved with the same asymptotic additive error

by using two counters to separately count increments and decrements. We present the following result based on Jain et al. (2023b) which uses the binary tree mechanism of Dwork et al. (2010); Chan et al. (2011) while leveraging Gaussian noise and zCDP. The result presented here is a special case of Theorem 5.4 of their work with $d = 1$.

**Theorem 2.1** (Gaussian Binary Tree Mechanism). *Let $\rho > 0$ be the privacy parameter. Consider a stream of updates $b_1, \ldots, b_T$ where $b_i \in \{-1, 0, 1\}$. There exists a randomized algorithm $\mathcal{A}(b_1, \ldots, b_T)$ which outputs estimates $\hat{y}_1^i, \ldots, \hat{y}_T^i$ with the following guarantees:*

- *Let two inputs $b_1, \ldots, b_T$ and $b_1', \ldots, b_T'$ be neighboring if and only if there exists a $j \in [T]$ such that $b_j \neq b_j'$ and $b_t = b_t'$ for all $t \neq j$. Under this neighboring definition, $\mathcal{A}$ preserves $\rho$-zCDP.*

- *For $t \in [T]$, let $y_t = \sum_{i=1}^t b_t$. Then, with probability $1 - 1/poly(T)$,*

$$\max_{t \in [T]} |y_t - \hat{y}_t| \leq O\big(\log^{1.5}(T)/\sqrt{\rho}\big).$$

- *$\mathcal{A}$ can be implemented as a streaming algorithm using $O(\log(T))$ words of space.*

## 2.3 OTHER PRELIMINARIES

We use a version of the Johnson-Lindenstrauss lemma with Rademacher random variables.

**Lemma 2.3** (Rademacher Johnson-Lindenstrauss Achlioptas (2003)). *Let $x \in \mathbb{R}^n$ and let $A$ be an $m \times n$ matrix with i.i.d. random entries $a_{i,j}$ with $\mathbf{Pr}[a_{i,j} = -1/\sqrt{m}] = \mathbf{Pr}[a_{i,j} = 1/\sqrt{m}] = 1/2$. Assume that $m \geq C \log(1/\delta)/\alpha^2$ for a sufficiently large constant $C$. Then $\mathbf{Pr}\big[(1-\alpha)\|x\|_2^2 \leq \|Ax\|_2^2 \leq (1+\alpha)\|x\|_2^2\big] \geq 1 - \delta$.*

Throughout the paper, we instantiate various hash functions. In Section 3, only pairwise independence is needed. In Sections 4 and 5, we do not optimize for randomness and note that $O(\log T)$-wise independence suffices.

## 3 CONTINUAL DISTINCT ELEMENTS VIA MINHASH

Without loss of generality, assume that $n$ is a power of two with $n = 2^K$ for $K \in \mathbb{N}$ (we can always artificially increase the universe size to achieve this). Consider a hash function $h : [n] \to [n]$. Given an integer $a \in [n]$, we define $\texttt{lsb}(a) \in \{0, \ldots, \lfloor \log n \rfloor\}$ to be the (zero-indexed) index of the least significant non-zero bit of $a$ in its standard binary representation. For example, $\texttt{lsb}(4) = 2$. Consider the $(K + 1)$-dimensional, 0-indexed vector $f_t$ where, for any $k \in \{0, \ldots, K\}$,

$$f_t[k] = \sum_{i=1}^t \mathbb{1}[\texttt{lsb}(h(a_i)) = k] \cdot s_i.$$

This quantity $f_t[k]$ is the sum of frequencies of distinct elements at time $t$ in the stream with $k$ trailing zeros. For each $k$, we will maintain an estimate $\hat{f}_t[k]$ over all times $t \in [T]$ via $(K + 1)$ DP Continual Counters $C[0], \ldots C[K]$ (Theorem 2.1).

A classic estimator for distinct elements is to consider the index of the largest non-zero bit.[4] The non-private template is to identify the largest $\ell \in \{0, ..., K\}$ such that $f_t[\ell] > 0$ and report $\hat{D} = 2^\ell$. The probability that a given element $a$ has $\texttt{lsb}(a) = k$ is equal to $2^{-(k+1)}$, so among $D$ elements, we roughly expect to see at least one element with $\texttt{lsb} \approx \log(D)$ and no elements with $\texttt{lsb} \gg \log(D)$.

The challenge with implementing this estimator with differential privacy is that a single change to the stream can possibly change the identity of the largest non-empty entry in $f_t$ for many timesteps $t$. By

---

[4]While this is a common technique for distinct element estimation, perhaps the most natural is to hash stream elements uniformly at random into [0,1] and use 1/(minimum hashed value) as an estimator for distinct elements. It is not clear if this technique is amenable to privatization. A naive calculation would upper bound the sensitivity of the minimum hashed value by O(1) (since the value is always in the interval [0, 1]), but this is not strong enough to be useful in the continual setting as the minimum hashed value could change often over the course of the stream as the result of a single event-level change.

using continual counters, we can only guarantee that $\hat{f}_t[k]$ is approximated up to $\tau = \text{polylog}(T)$ additive error. So, instead of identifying the largest non-empty entry of $f_t$, we try to identify the largest entry of $f_t$ which exceeds this noise threshold $\tau$. If stream elements were bounded to have at most constant frequency, this would immediately yield an algorithm with error $(O(1), \tau)$.[5] However, stream elements could have frequencies larger than $\tau$. Therefore, we do not know if the bucket we find has count $f_t[\ell] > \tau$ because (a) $\ell \approx \log(D_t/\tau)$ and approximately $\tau$ elements of constant frequency have $\texttt{lsb} = \ell$ or (b) $\ell \approx \log(D)$ and a single element of frequency more than $\tau$ has $\texttt{lsb} = \ell$. This is the source of the $O(\tau)$ multiplicative error.

---

**Algorithm 1: MinHash Subroutine**

**Input:** Privacy parameter $\rho$, stream length $T$, domain size $n = 2^K < \text{poly}(T)$.

1. Initialize $\rho'' = \rho/2$.

2. Initialize a pairwise independent hash function $h : [n] \to [n]$.

3. For $k \in \{0, \ldots, K\}$, initialize a DP Continual Counter (Theorem 2.1) $C[k]$ with privacy parameter $\rho''$. Let $\tau = \Theta(\log^{1.5}(T)/\sqrt{\rho})$ be an upper bound on the $(1 - 1/\text{poly}(T))$ quantile of the maximum additive error of a counter over all timesteps $t \in [T]$.

4. When a stream update $(a_t, s_t)$ arrives,

    (a) $k \leftarrow \texttt{lsb}(h(a_t))$.

    (b) Update $C[k]$ with update $s_t$. Update all other counters $C[k']$ for $k' \neq k$ with update $0$.

    (c) Let $\hat{f}_t[k']$ be the current estimate of $C[k']$ for all $k' \in \{0, \ldots, K\}$. Let $\ell$ be the largest index such that $\hat{f}_t[\ell] > \tau$. If no such index exists, let $\ell \leftarrow 0$.

    (d) **Output:** $\hat{D}_t \leftarrow 2^\ell$.

---

**Algorithm 2: MinHash Estimator**

**Input:** Privacy parameter $\rho$, stream length $T$, domain size $n = 2^K < \text{poly}(T)$.

1. Let $m = \Theta(\log T)$

2. Initialize $\rho' = \rho/m$.

3. Run $m$ copies of Algorithm 1 in parallel with privacy parameter $\rho'$.

4. For each timestep $t \in [T]$,

    (a) For $j \in [m]$, let $\hat{D}_t^j$ be the estimate of the $j$th subroutine at time $t$.

    (b) Report estimate $\text{median}\left(\hat{D}_t^1, \ldots, \hat{D}_t^m\right)$.

---

**Theorem 3.1.** *Algorithm 2 is $\rho$-zCDP and uses space $O(\log n \cdot \log^2(T))$. On strict turnstile streams, if $\rho < O(\log^4 T)$, Algorithm 2 solves continual distinct elements estimation with $(O\left(\log^2(T)/\sqrt{\rho}\right), O\left(\log^2(T)/\sqrt{\rho}\right))$ error with probability $1 - 1/\text{poly}(T)$.*

Note that the condition on $\rho$ not being too large relative to $T$ is mild as the privacy parameter is often chosen to be a small constant. The proof of this theorem is given in Appendix C.

## 4 CONTINUAL DISTINCT ELEMENTS VIA DOMAIN REDUCTION

We now present another algorithmic technique which guarantees a $\text{polylog}(T)$ multiplicative approximation to the number of distinct elements in a private turnstile stream (see Section 2). The main theorem of this section is Theorem 4.1. While the bounds of Theorem 4.1 are quantitatively

---

[5]Under this promise, the algorithm would also succeed with high probability via a Chernoff bound without the need for independent repetitions.

worse than of our main theorem, Theorem 3.1, this section serves to demonstrate new ideas for the private continual estimation setting, which may be of independent interest. Additionally, they apply to the general turnstile model while Theorem 3.1 applies only to the strict turnstile model where frequencies are non-negative at all times.

---

**Algorithm 3: Domain Reduction Estimator**

**Input:** Privacy parameter $\rho$, stream length $T$

1. Let $C, C' \geq 1$ be sufficiently large constants.

2. Set $\rho' = \rho/(C \log^2 T)$.

3. For $1 \leq i \leq \log T$ and $1 \leq j \leq C \log T$, construct independent functions $f^i_{g_j, h_j} : [n] \to [2^i]$ as in Lemma D.1.

4. For all $i, j$ and all time steps $t \in [T]$, compute a coordinate-wise frequency estimate $\tilde{f}^i_{g_j, h_j}(x_t)$ of $f^i_{g_j, h_j}(x_t)$ in the streaming continual release model, each of which is $\rho'$-zCDP. ▷ *This guarantees $\|f^i_{g_j, h_j}(x_t) - \tilde{f}^i_{g_j, h_j}(x_t)\|_\infty \leq \tau$ for all $i, j, t$ with failure probability $\beta$ where $\tau = O(\log^{1.5}(T)/\sqrt{\rho'})$ via Theorem 2.1.*

5. For all $i$ and all time steps $t \in [T]$, let $\hat{F}^i(t) \in \mathbb{R}^{2^i}$ denote the (coordinate-wise) median of $\tilde{f}^i_{g_1, h_1}(x_t), \ldots, \tilde{f}^i_{g_{100 \log T}, h_{100 \log T}}(x_t)$. ▷ *Note $i$ is fixed.*

6. For every time step $t \in [T]$, let $i^*(t)$ be the largest value such that all $\|\hat{F}^{i^*}(t)\|_\infty \geq C'\tau$.

7. **Return:** $2^{i^*(t)}$ as an estimate of $\|x_t\|_0$ at time $t \in [T]$.

---

**Theorem 4.1.** *Algorithm 3 satisfies $\rho$-zCDP. It solves the continual distinct elements estimation with*

$$\left(O(\log^{10}(T)/\rho^2), O(\log^{10}(T)/\rho^2)\right)$$

*error with probability $1 - 1/poly(T)$. The algorithm uses polynomial in $T$ space and works in the general turnstile model.*

The same techniques allow us to show a reduction from multiplicative error to additive error (in the domain size), showing that an existence of an algorithm (for our distinct elements problem) with *sublinear* additive error in the domain size implies an algorithm which guarantees *multiplicative* approximations arbitrarily close to 1. In the theorem below, recall that $n$ is the size of the universe. We also restrict $polylog(T) \geq \rho \geq 0$ in the theorem statement below, since we will use the algorithm of Theorem 3.1.

**Theorem 4.2.** *Suppose there exists a constant $1 > c_1 \geq 0$ such that there is a $\rho$-zCDP algorithm $\mathcal{A}$ for the continual distinct elements problem which has additive error $n^{c_1}/poly(\rho)$ and is correct with probability $1 - 1/poly(T)$ for all time steps. Then for any $\eta \in (0, 1)$, there exists another $\rho$-zCDP algorithm $\mathcal{A}'$ (for the continual distinct elements problem) which achieves a multiplicative error $1 + \eta$, additive error $poly(\log(T), 1/\eta, 1/\rho)$ and is also correct with probability $1 - 1/poly(T)$.*

At a high-level, Algorithm 3 is giving a dimensionality reduction map which preserves the number of distinct elements up to poly-logarithmic multiplicative error. Unlike traditional dimensionality reduction however, we are not necessarily interested in reducing the dimension of the frequency vector (which is the size of the universe). Rather, we exploit a different property of dimensionality reduction which is that it increases the value of the coordinates in the reduced dimension. This is useful from a DP perspective, as large coordinates can be detected via continual counting.

The proofs of these theorems are given in Appendix D.

## 5 Continual $F_2$ estimation

We next consider the problem of $F_2$ estimation. To recap, the stream consists of $T$ stream elements $S = (a_t, s_t)_{t \in [T]}$ where $a_t \in [n]$ and $s_t \in \{-1, 0, 1\}$. For $t \in [T]$ and $j \in [n]$, we define $x_t[j] = \sum_{i \leq t : a_i = j} s_i$. We assume the strict turnstile model where $x_t[j] \geq 0$ always. Define $x^{(t)}$

to be the $n$-dimensional vector $(x_t[j])_{j\in[n]}$. In this section our interest is to estimate the second moment $F^{(t)}(S) = \sum_{j\in[n]}(x_t[j])^2 = \|x^{(t)}\|_2^2$ at any point of time $t$.

We first prove a simple lower bound using standard techniques based on the sensitivity of the second moment when a single stream element is changed. Note, that the lower bound works even against non-streaming algorithms that see the entire stream up-front.

**Lemma 5.1.** *Let $\mathcal{S}$ be the set of all streams $S = (a_i, s_i)_{i\in[T]}$. Let $\mathcal{M} : \mathcal{S} \to R$ be an $(\epsilon, \delta)$-DP protocol. Then there exists some $S_0 \in \mathcal{S}$ such that*

$$\mathbf{Pr}\Big[|\mathcal{M}(S_0) - F_2^{(T)}(S_0)| \geq T - 1\Big] \geq \frac{1-\delta}{e^\varepsilon + 1}.$$

Thus, the additive error incurred by any DP $F_2$ estimation algorithm is $\Omega(T)$.

*Proof of Lemma 5.1.* Let $S$ be the stream where all stream elements are $(a_i, s) = (1, 1)$ and then $S'$ be the neighboring stream which is identical to $S$ except that $(a_T, s_T) = (2, 1)$. Then, $F_2^{(T)}(S) = T^2$ and $F_2^{(T)}(S') = (T-1)^2 + 1^2$ so that, $F_2^{(T)}(S) - F_2^{(T)}(S') = 2T - 2$. Define $\lambda = \frac{F_2^{(T)}(S)+F_2^{(T)}(S')}{2}$ and $p = \frac{1-\delta}{e^\varepsilon+1}$. If $\mathbf{Pr}[\mathcal{M}(S) \leq \lambda] \geq p$, the result follows with $S_0 = S$, so assume that $\mathbf{Pr}[\mathcal{M}(S) \leq \lambda] < p$. Then, by differential privacy,

$$\mathbf{Pr}[\mathcal{M}(S') > \lambda] \geq \frac{\mathbf{Pr}[\mathcal{M}(S) > \lambda] - \delta}{e^\varepsilon} > \frac{1 - p - \delta}{e^\varepsilon} = p,$$

so the result follows with $S_0 = S'$. □

Notice that the lower bound instance used to prove the above lemma has a single element $i \in [n]$ with frequency $\Omega(T)$. However, for such an input $S$, the value $F_2(S)$ is of the order $\Omega(T^2)$ which is much larger than the additive error. Motivated by this observation, we ask if it is possible to obtain better additive error of our estimates if we allow a small *multiplicative* error of $(1 + \alpha)$. In this section, we show that for any small constant $\alpha$, it is possible to obtain additive error polylog$(T)$. Note that Epasto et al. (2023) obtain a qualitatively similar bound, but in the restricted setting of insertion-only streams.

At a high-level, our algorithm is inspired by the classic AMS sketch for $F_2$ estimation in standard streaming Alon et al. (1996), and uses the classic Johnson-Lindenstrauss (JL) lemma Lemma 2.3 to reduce the domain size from $[n]$ to polylog$(T)$. By using a JL construction which utilizes Rademacher random variables, we are able to approximately (and privately) track the frequencies of the domain elements *after* applying the JL lemma via continual counting (see Theorem 2.1). Composition (Lemma 2.1) allows us to simultaneously release the frequencies in the reduced domain with a low-additive error overhead since the *size* of the domain has reduced significantly. Altogether, we show that the frequencies in the reduced domain can all be simultaneously estimated up to polylog$(T)$ additive error, which implies our desired error bound. The $1 + \alpha$ multiplicative error is incurred by the JL step itself.

---

**Algorithm 4: $F_2$ Estimator**

**Input:** Privacy parameter $\rho$, stream length $T$, domain size $n$, approximation parameter $\alpha > 0$.

1. Let $\alpha_0 = \alpha/5$.
2. Let $m = C_1(\log T)/\alpha_0^2$ with $C_1$ sufficiently large according to Lemma 2.3.
3. Initialize random $m \times n$ matrix $A$ where the $A_{ij}$ are i.i.d with $\mathbf{Pr}[A_{ij} = 1/\sqrt{m}] = \mathbf{Pr}[A_{ij} = -1/\sqrt{m}] = 1/2$.
4. For $i \in \{1, \ldots, m\}$, initialize a DP Continual Counter (Theorem 2.1) $C[i]$ with privacy parameter $\rho' = \rho/m$.
5. When a stream element $(a_t, s_t)$ arrives,
    (a) Let $j = a_t \in [n]$.
    (b) For each $i \in [m]$, update $C[i]$ with $\sqrt{m}A_{ij}s_j \in \{-1, 0, 1\}$.
6. For each timestep $t \in [T]$,
    (a) Report estimate $\frac{1}{m}\sum_{i=1}^{m} C[i]^2$.

**Theorem 5.1.** *Let $0 < \alpha < 1$ and privacy parameter $\rho > 0$. There exists an $\rho$-zCDP algorithm which for all $t \in [T]$ provides an estimate $\hat{F}_2^{(t)}$ of $\|x^{(t)}\|_2^2$ such that with high probability in $T$,*

$$|\hat{F}_2^{(t)} - \|x^{(t)}\|_2^2| \leq \alpha\|x^{(t)}\|_2^2 + O\left(\frac{(\log T)^4}{\alpha^3\rho}\right),$$

*Moreover, the algorithm can be implemented in the streaming model where it uses $O((\log T)^2/\alpha^2)$ words of memory.*

Note again that to obtain an algorithm which satisfies $(\varepsilon, \delta)$-DP, it suffices to pick $\rho = O(\varepsilon^2/\log(1/\delta))$. The algorithm behind this result is Algorithm 4, and the proof is given in Appendix E.

## ACKNOWLEDGMENTS

The authors thank anonymous ICLR 2026 reviewers for helpful feedback. The authors thank Xin Lyu and A. R. Sricharan for discussions on their concurrent works.

Anders Aamand was supported by the VILLUM Foundation grant 54451. Justin Y. Chen was supported by the NSF TRIPODS program (award DMS-2022448) and NSF Graduate Research Fellowship under Grant No. 1745302.

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

# A  PRIOR WORK

In this section, we describe existing results on $(\varepsilon, \delta)$-DP continual distinct elements and $F_2$ estimation. Beyond directly related prior work, there are several works which investigate other streaming problems (not necessarily with continual release) achieving small space along with privacy guarantees Smith et al. (2020); Zhao et al. (2022); Pagh & Thorup (2022); Janos Lebeda & Tetek (2024); Qiu & Yi (2025). For simplicity, we ignore dependencies on the privacy parameters in the error bounds in following presentation.

**Item-Level Privacy**  The recomputation technique of Jain et al. (2023b) can be used to get an algorithm for turnstile streams which achieves error $(1, \tilde{O}(T^{1/3}))$ by recomputing a private estimate of the number of distinct elements every $T^{1/3}$ timesteps and outputting the most recent estimate at every $t \in [T]$. This algorithm uses space which is linear in either $T$ or $n$ as the exact distinct element count must be maintained throughout the stream.[6]

Jain et al. (2023a) initiated the specific study of the turnstile continual distinct elements problem. This and several follow-up works do not generally improve upon $T^{1/3}$ additive error but introduce error bounds parameterized by statistics of the stream instance. Therefore, for certain instances, these bounds may improve upon the worst-case. This work gives an algorithm achieving additive error $\tilde{O}(\sqrt{w})$ where $w$ is the *flippancy* of the stream: the maximum number of times any one element switches between zero frequency and positive frequency. This algorithm requires $O(T)$ space in order to estimate the flippancy. The authors show that $\tilde{\Omega}(\max\{\sqrt{w}, T^{1/3}\})$ purely additive error is required. Henzinger et al. (2024a) consider a quantity $K$ which is the *total flippancy*: the total number of times any element switches between zero and positive frequency. They show that $\tilde{\Theta}(K^{1/3})$ purely additive error is achievable and required.

**Event-Level Privacy**  Event-level privacy is strictly weaker than item-level privacy. Therefore, algorithms for item-level privacy also apply to event-level privacy, while lower bounds for event-level privacy imply lower bounds for item-level privacy.

Jain et al. (2023a) show that for event-level privacy, a weaker lower bound $\tilde{\Omega}(\max\{\sqrt{w}, T^{1/4}\})$ purely additive error is required. This leaves open an interesting gap between the upper bound of $T^{1/3}$ (which is achievable with item-level privacy) and lower bound of $T^{1/4}$. Henzinger et al. (2024a) show that even for event-level privacy, $\tilde{\Theta}(K^{1/3})$ is the best possible dependence on the total flippancy for purely additive error.

Cummings et al. (2025) give an algorithm achieving $\left((1 + \eta, \tilde{O}_\eta(\sqrt{v})\right)$ error where $v$ is the *occurrency*: the maximum number of times any given element is updated in the stream. While $v > w$, their algorithm has the benefit of using space $\tilde{O}_\eta(T^{1/3})$. While this work allows for multiplicative error, their goal was simply to achieve the existing bound of $T^{1/3}$ with low space (which necessitates some multiplicative error). In this work, we show that allowing (more) multiplicative error allows for significantly better additive error, and we achieve polylogarithmic as opposed to polynomial space.

To our knowledge, there has been no prior work specifically studying continual estimation of other frequency moments such as $F_2$ in *turnstile* streams.

**Insertion-Only Streams and the "Likes" Model**  When deletions are not allowed (or limited), the problem of continual distinct elements becomes significantly easier.

Epasto et al. (2023) study the problem of continual release of frequency moments (including the number of distinct elements, which is equivalent to the zeroth frequency moment) in the *insertion-only* model where stream updates must be increments with $s_t = 1$. They give an $(\varepsilon, 0)$-DP algorithm achieving $\left((1 + \eta, O_{\varepsilon,\eta}(\log^2(T)))\right)$ error algorithm using space $\text{poly}(\log T, 1/\eta, 1/\varepsilon)$. The authors

---

[6]It is tempting to make a sublinear space version of Jain et al. (2023b) by applying the recomputation technique to a multiplicative approximation to the number of distinct elements at any time using a standard streaming algorithm. However, this idea does not straightforwardly work as such approximation algorithms blow up the sensitivity.

also study $F_2$ estimation in insertion-only streams. They achieve $\left(1 + \eta, \tilde{O}_\eta(\log^7(T))\right)$ error using space $O(\log^2(T)/\eta^2)$.

Henzinger et al. (2024a) consider the "likes" model in which elements may only switch from having frequency zero or one. This is an intermediate model between insertion-only and strict turnstile streams. In this setting, polylog($T$) additive error is necessary and achievable.

**Concurrent Works of Aryanfard et al. (2025); Andersson et al. (2026); Epasto et al. (2026)** Three concurrent works study various related aspects of private continual estimation. There is no direct overlap with our results, though combined, their and our results significantly expand the landscape of private turnstile estimation.

The work of Aryanfard et al. (2025) studies mixed multiplicative and additive error in continual settings with a focus on graph problems. They show that hardness for purely additive error for several fundamental graph problems in the continual setting hold for *insertion-only* as well as turnstile streams. On the other hand, they show improved bounds for several insertion-only problems with mixed multiplicative and additive error including graph problems and simultaneous norm estimation. They do not provide any new upper bounds for turnstile streams, which is the focus of our work. Finally, they show that for continual estimation of some graph statistics in the *item-level* privacy setting, the product of multiplicative and additive errors must be large. Their lower bounds (either via a reduction from distinct elements to the EdgeCount problem or by their lower bound on 1-Way-Marginals combined with the reduction of Jain et al. (2023a)) extend to distinct elements: the product of multiplicative and additive errors $\alpha\beta$ must be essentially $T^{1/3}$.[7] This means that an analogue of our upper bounds, which hold for event-level DP, cannot hold for item-level DP.

The work of Andersson et al. (2026) improves constant factors in the additive error for continual release of distinct elements, degree counts, and triangle counts. Their main contribution is to adapt matrix factorization techniques which improve constant factors for continual counting (e.g., Henzinger et al. (2023)) to replace the binary tree mechanism in existing approaches such as in Jain et al. (2023a). In comparison, our results give asymptotically better additive error at the price of some multiplicative error.

The work of Epasto et al. (2026) proves space lower bounds against algorithms achieving results similar to those of Jain et al. (2023a); Cummings et al. (2025) in the item-level setting. In particular, they show that any algorithm achieving error $(1.1, o(T^{1/3}))$ (where the improvement over $T^{1/3}$ error may be due to instance-specific occurrency statistics as in Cummings et al. (2025)) must use space close to $T^{1/3}$. We show that it is possible in a different setting to simultaneously achieve small space and small additive error, specifically, in the event-level privacy setting when the multiplicative error is significantly more than $1.1$.[8]

---

[7]We thank the authors of Aryanfard et al. (2025) for pointing out that their lower bounds can be extended to the distinct elements problem.

[8]From discussions with the authors of Epasto et al. (2026), it seems likely that their lower bound can be extended to large multiplicative error.

# B    OPEN PROBLEMS

We discuss several open problems exploring the trade-offs between multiplicative and additive errors in private continual estimation settings.

**Better dependence on $n$ and $T$ for counting distinct elements.**    Perhaps the main open question of our paper is whether one can obtain better bounds for counting distinct elements under differential privacy with purely additive error. Without any assumptions on the input stream, there is a polynomial gap between the upper bound of $\tilde{O}(T^{1/3})$ and the lower bound of $\Omega(T^{1/4})$. Moreover, the lower bound holds only with some restrictions on the values of $n$ and $T$, namely under the assumption that $T \leq n^2$. For general values of $n$ and $T$, the lower bound is $\Omega(\min(T^{1/4}, n^{1/2}))$ which is never stronger than a lower bound of $n^{1/2}$. Note that when counting distinct elements over a domain of size $n$, it is trivial to obtain error $n$. An interesting question towards resolving the optimal dependence on the parameters in the additive error is whether there exists an algorithm with error $o(n)$ when $T$ is a large polynomial in $n$. This is especially interesting in light of our following result.

**Theorem B.1** (Informal version of Theorem 4.2). *Any $(\varepsilon, \delta)$-DP algorithm for the number of distinct elements in the continual release setting with purely additive error $n^{0.99}$ can be converted to another differentially private algorithm with $(1 + \eta)$-multiplicative and $poly(\log T, 1/\eta, 1/\varepsilon, \log(1/\delta))$ additive error for any constant $\eta > 0$.*

Alternatively, this reduction could be used as a path towards proving lower bounds against $o(n)$ purely additive error.

**Constant multiplicative approximation to distinct elements with small additive error.**    Our results show that we can avoid the large additive error of the lower bounds, if we allow for a multiplicative error of $polylog(T)$ for counting distinct. It is an interesting question whether there exists an algorithm with *constant* multiplicative error and small (e.g., polylogarithmic) additive error. Our techniques based on continual counting seems to reach a natural barrier here, which arises from the fact that the counters cannot distinguish between whether the count of a bucket comes from a single highly frequent element, or from many infrequent elements, each having frequency 1, say.

**Tradeoff between multiplicative and additive error.**    Taking a step further, what is the correct tradeoff between multiplicative and additive error for counting distinct elements? If a distinct elements algorithm with $(1 + \eta)$ multiplicative approximation and small additive error exists, what is the dependence on $\eta$ in the additive error? A similar question can be asked for $F_2$ estimation, where our current algorithm has $(1 + \eta)$ multiplicative approximation and $\tilde{O}(1/\eta^3)$ additive error. More broadly, it is interesting to explore tradeoffs between multiplicative and additive error in other settings for private continual estimation. A potential direction is to adapt the lower bound techniques in the concurrent work of Aryanfard et al. (2025).

**Triangle Counting.**    Recently, Raskhodnikova & Steiner (2024) considered differentially private triangle counting in dynamic graphs. Their paper contains several results, but most relevant to our setting is an algorithm for counting triangles in a graph with bounded additive error depending on $T$ and the number of vertices of the graph. The algorithm and analysis of this problem mirror continual distinct elements estimation, and it is interesting to see whether we can obtain better additive error guarantees if we are allowed a small multiplicative error.

# C  PROOFS FOR SECTION 3

We first prove intermediate lemmas about the error and privacy of Algorithm 1.

**Lemma C.1.** *Given privacy parameter $\rho$ as input, Algorithm 1 preserves $\rho$-zCDP.*

*Proof.* Consider the algorithm $\mathcal{A}$ which, at every timestep $t$, reports the $K + 1$ dimensional vector $\hat{f}_t$. Note that Algorithm 1 is simply a post-processing of this hypothetical algorithm. So, it suffices to consider the privacy of $\mathcal{A}$. Consider two neighboring data streams $X = (a_1, s_1), \ldots, (a_T, s_T)$ and $X' = (a'_1, s'_1), \ldots, (a'_T, s'_T)$ which are the same for all timesteps except for some $i \in [T]$ where $(a_i, s_i) \neq (a'_i, s'_i)$.

Note that the randomness of $h$ is independent of the randomness used for privacy of all counters $C[k]$. Fix $h$ and let $k_1 = \mathtt{lsb}(h(a_i))$ and $k_2 = \mathtt{lsb}(h(a'_i))$. The distribution over $(\hat{f}_t[k])_{t \in [T], k \in \{0, \ldots, K\} \setminus \{k_1, k_2\}}$ is the same for $\mathcal{A}$ run on $X$ and $X'$. We will proceed by cases

Consider the case where $k_1 \neq k_2$, and consider the counters $C[k_1]$ and $C[k_2]$. The updates are the same to these counters under $X$ and $X'$ at all timesteps other than time $i$. At time $i$, under $X$, $C[k_1]$ receives update $s_i$ and $C[k_2]$ receives update $0$ while under $X'$, $C[k_1]$ receives update $0$ and $C[k_2]$ receives update $s'_i$. This exactly corresponds to the neighboring datasets relation for DP Continual Counting in Theorem 2.1. As $C[k_1]$ and $C[k_2]$ are $\rho''$-zCDP with $\rho'' = \rho/2$, by composition (Lemma 2.1), $\mathcal{A}$ is $\rho$-zCDP.

Consider the case where $k = k_1 = k_2$. The updates to $C[k]$ are the same for all timesteps except at the $i$th timestep where the update is $s_i$ under $X$ and $s'_i$ under $X'$. By the privacy of $C[k]$ given in Theorem 2.1, $\mathcal{A}$ satisfies $(\rho/2)$-zCDP. $\square$

**Lemma C.2.** *Consider any fixed timestep $t \in [T]$. If $\rho = O(\log^3 T)$ and the stream is in the strict turnstile model, then the output of Algorithm 1 satisfies*

$$D_t/6\tau \leq \hat{D}_t \leq 4D_t + 1$$

*with probability $2/3$.*

*Proof.* By the definition of $\tau$, with probability $1/\mathrm{poly}(T)$,

$$\tau \geq \max_{t \in [T], k \in \{0, \ldots, K\}} \left| f_t[k] - \hat{f}_t[k] \right|.$$

For the rest of the analysis, assume that this upper bound holds.

Consider any $i \in \{0, \ldots, K\}$. For a given element $a \in [n]$, $\mathbf{Pr}_h[\mathtt{lsb}(a) = i] = 2^{-(i+1)}$ and $\mathbf{Pr}_h[\mathtt{lsb}(a) \geq i] = 2^{-i}$.

We will first show that the estimate $\hat{D}_t$ cannot be too small. It is always the case that $\ell \geq 0$, so $\hat{D}_t \geq 1$. Note that if $D_t \leq 6\tau$, then $\hat{D}_t \geq D_t/6\tau$ by default.

Consider the case that $D_t > 6\tau$, and let $i = \lfloor \log(D_t/6\tau) \rfloor$. Let $Z_i$ be a random variable for the number of elements in $S_t$ with least significant bit equal to $i$. Under full randomness, $Z_i$ would be distributed as $\mathrm{Bin}(D_t, 2^{-(i+1)})$. Then, $\mu = \mathbf{E}[Z_i] \in [3\tau, 6\tau]$ and $\mathbf{Var}[Z_i] \leq \mu$. By Chebyshev's inequality,

$$\mathbf{Pr}_h[Z_i \leq 2\tau] \leq \mathbf{Pr}[|Z_i - \mu| \leq \mu - 2\tau] \leq \frac{\mathbf{Var}[Z_i]}{(\mu - 2\tau)^2} \leq \frac{6}{\tau}.$$

This quantity is upper bounded by $1/100$ as long as $\tau$ is a sufficiently large constant which is implied by $\rho = O(\log^3(T))$.

With probability at least $99/100$, there exists an index $i \geq \lfloor \log\left(\frac{D_t}{6\tau}\right) \rfloor$ such that $Z_i > 2\tau$. As all elements in $S_t$ have frequency at least $1$ (this is where we use the strict turnstile model) and by the additive error given by Theorem 2.1, $\hat{f}_t[i] > \tau$. As this is a valid choice for $\ell$, $\ell \geq \lfloor \log\left(\frac{D_t}{6\tau}\right) \rfloor$, so $\hat{D}_t = 2^\ell \geq \frac{D_t}{6\tau}$.

Next we will show that $\ell$ cannot be too large. Any empty bucket with $Z_i = 0$ will have $\hat{f}_t[i] \leq \tau$. Consider the case where $D_t > 0$ and let $j = \lceil \log(4D_t) \rceil$. Then,

$$\mathbf{Pr}_h \left[ \sum_{i=j}^{K} Z_i = 0 \right] = \left(1 - 2^{-j}\right)^{D_t} \geq 1 - 2^{-j} D_t \geq 3/4.$$

Under this event, $\ell < j$, so $\hat{D}_t \leq 4D_t$. Note that if $D_t = 0$, then $\sum_{j=0}^{K} Z_i = 0$ and $\hat{D}_t = 1$.

Union bounding over all events, with probability $1 - 1/\text{poly}(T) - 1/100 - 1/100 - 1/4 \geq 2/3$, the following holds:

$$D_t/6\tau \leq \hat{D}_t \leq 4D_t + 1.$$

$\square$

*Proof of Theorem 3.1.* We will separately prove that the subroutine is private, accurate, and has bounded space.

**Privacy**   Algorithm 2 is a post-processing of the $m$ copies of the subroutine Algorithm 1. By the privacy of the subroutine (Lemma C.1) and composition (Lemma 2.1), Algorithm 2 satisfies $m\rho' = \rho$-zCDP.

**Error**   Lemma C.2 bounds the error, for each timestep $t$, of each estimator $\hat{D}_t^j$ by

$$D_t/6\tau \leq \hat{D}_t \leq 4D_t + 1$$

where $\tau = \Theta\big(\log^{1.5}(T)/\sqrt{\rho'}\big) = \Theta\big(\log^2(T)/\sqrt{\rho}\big)$. This error bound holds with probability $2/3$ under the condition that $\rho' = O(\log^3(T))$ which is equivalent to $\rho = O(\log^4(T))$.

By a standard argument, the median estimator will violate this error bound with probability $\exp(-\Omega(m)) = 1/\text{poly}(T)$ via a Hoeffding bound. The result follows by union bounding over all $T$ timesteps.[9]

**Space**   Each subroutine Algorithm 1 maintains $K + 1 = O(\log n)$ DP Continual Counters which each require $O(\log T)$ space due to Theorem 2.1. In the error analysis of Lemma C.2, we require that the hash function $h : [n] \to [n]$ satisfies certain pseudorandomness conditions. Specifically, we use the first and second moments of the random variables $Z_i$ under full randomness, which is guaranteed by a pairwise independent hash family. Such a hash function can be stored in $O(1)$ words of space Carter & Wegman (1979). So, a single subroutine uses $O(\log n \cdot \log T)$ words of space. The overall algorithm with $m$ copies of the subroutine uses space $O(\log n \cdot \log^2(T))$.   $\square$

---

[9]Technically, we prove the stronger result that the algorithm satisfies $(O\big(\log^2(T)/\sqrt{\rho}\big), 1)$ error. We choose to present the result as $(O\big(\log^2(T)/\sqrt{\rho}\big), O\big(\log^2(T)/\sqrt{\rho}\big))$ as we only get non-trivial multiplicative approximation of the number of distinct elements when it exceeds $O\big(\log^2(T)/\sqrt{\rho}\big)$.

# D  PROOFS FOR SECTION 4

The proofs of Theorems 4.1 and 4.2 are based on a series of *domain reduction* lemmas, which allow us to reduce the size of the universe, while approximately preserving the number of distinct elements of the stream. The first lemma, Lemma D.1, gives anti-concentration bounds for the frequency of items when we randomly hash the domain to a smaller universe. It shows that if the reduced domain size is smaller than the number of non-zero entries of the frequency vector by polylogarithmic factors, then all the non-zero frequencies in the reduced domain are "sufficiently large." Conversely, Lemma D.2 shows that if the reduced domain is sufficiently larger than the number of non-zero entries, then any single coordinate in the reduced domain is likely to be zero.

These two lemmas form the foundation of the proof of Theorem 4.1. At a high level, we can reduce estimating the number of distinct elements to frequency estimation, which can be tracked up to polylogarithmic error via Theorem 2.1. To summarize, this is because if we reduce the domain to the "right size" (comparable to $\|x_t\|_0$ up to polylogarithmic factors), all domain elements will have *large* frequencies, and hence can be detected via Theorem 2.1.

Finally, the third domain reduction lemma, Lemma D.3, shows that the $\ell_0$ norm of a vector is preserved very precisely, if we again map the domain to a suitably larger domain. This is the main technical tool in Theorem 4.2.

**Lemma D.1** (Domain Reduction Lemma 1). *Suppose $n$ is sufficiently large. Let $h : [n] \to [m]$ and $g : [n] \to \{\pm 1\}$ be random functions. For a vector $x \in \mathbb{Z}^n$, define the mapping $f_{g,h}(x) \in \mathbb{Z}^m$ as*

$$\forall i \in [m], \quad f_{g,h}(x)_i = \left| \sum_{j \in [n], h(j)=i} g(j) \cdot x_j \right|,$$

*with an empty sum denoting $0$. If $1 \le m \le \frac{\|x\|_0}{\ell}$ for some $\ell \ge 100 \log(n)$, then for every fixed $i \in [m]$, we have*

$$\mathbf{Pr}_{h,g}\left[ f_{g,h}(x)_i \ge \frac{\sqrt{\ell}}{1000} \right] \ge 0.97.$$

*Proof.* Fix an $i \in [m]$. Let $S_i = \{j \in [n] \mid j \in \mathrm{supp}(x), h(j) = i\}$ denote the elements in the support of $x$ that hash to the $i$th coordinate. Note that $|S_i| \sim \mathrm{Bin}\left(\|x\|_0, \frac{1}{m}\right)$. Noting that $\mathbf{E}_h[|S_i|] \ge \ell \ge 100 \log(n)$, we have that $|S_i| \ge \ell/2$ with probability at least $0.999$.

Now conditioned on $|S_i| \ge \ell/2$, we know that

$$\sum_{j \in S_i} g(j) \cdot x_j$$

is a sum of weighted Rademacher random variables, where each $|x_j| \ge 1$. Since $g$ is a random function, we can assume without loss of generality that $x_j \ge 1$. Thus, the Erdos-Littlewood-Offord bound Erdős (1945), again conditioned on $|S_i| \ge \ell/2$, implies that

$$\sup_{|I| \le 1} \mathbf{Pr}_g\left[ \sum_{j \in S_i} g(j) \cdot x_j \in I \right] \le \frac{50}{\sqrt{\ell}},$$

for any interval $I$ of length at most $1$. By a union bound over intervals,

$$\mathbf{Pr}_g\left[ \sum_{j \in S_i} g(j) \cdot x_j \in \left[ -\frac{\sqrt{\ell}}{1000}, \frac{\sqrt{\ell}}{1000} \right] \,\middle|\, |S_i| \ge \ell/2 \right] \le \frac{\sqrt{\ell}}{2000} \cdot \frac{50}{\sqrt{\ell}} \le \frac{1}{40}.$$

Therefore, we have

$$\mathbf{Pr}_{h,g}\left[ f_{g,h}(x)_i \ge \frac{\sqrt{\ell}}{1000} \right] \ge \mathbf{Pr}_g\left[ \sum_{j \in S_i} g(j) \cdot x_j \notin \left[ -\frac{\sqrt{\ell}}{1000}, \frac{\sqrt{\ell}}{1000} \right] \,\middle|\, |S_i| \ge \ell/2 \right] \cdot \mathbf{Pr}_h[|S_i| \ge \ell/2]$$

$$\ge 0.999(0.975) \ge 0.97,$$

as desired. $\qquad\qquad\square$

**Lemma D.2** (Domain Reduction Lemma 2). *Consider the same setting as Lemma D.1 but suppose $m \geq \ell\|x\|_0$. Then for every fixed $i \in [m]$, we have*

$$\mathbf{Pr}_{h,g}[f_{g,h}(x)_i \neq 0] \leq 0.01.$$

*Proof.* We know $|S_i| \sim \text{Bin}(\|x\|_0, \frac{1}{m})$, so $\mathbf{E}_h[|S_i|] \leq \frac{1}{\ell}$. Thus, the probability that $|S_i| \neq 0$ is at most $1/\ell \leq 0.01$ by Markov's inequality. In the case that $S_i = \emptyset$, it is clear that $f_{g,h}(x)_i = 0$, as desired. $\qquad\square$

**Lemma D.3** (Domain Reduction Lemma 3). *Suppose $n$ is sufficiently large and let $h : [n] \to [m]$ be a random function as in Lemma D.1. For a vector $x \in \mathbb{R}^n$, define the mapping $f_n(x) \in \mathbb{R}^m$ as*

$$\forall i \in [m], \quad f_h(x)_i = \sum_{j \in [n], h(j) = i} x_j,$$

*with an empty sum denoting $0$. If $m \geq \|x\|_0 \cdot \ell \cdot \ell'$ for some $\ell, \ell' \geq 1$, then*

$$\mathbf{Pr}_h\left[\|x\|_0 \geq \|f_h(x)\|_0 \geq \left(1 - \frac{1}{\ell}\right)\|x\|_0\right] \geq 1 - \frac{1}{\ell'}.$$

*Proof.* It is clear that $\|x\|_0 \geq \|f_h(x)\|_0$ holds deterministically, as $f_h$ may have collisions. For the other direction, for $j \in \text{supp}(x)$, let $X_j$ denote the indicator variable that no other $j' \in \text{supp}(x)$ satisfies $h(j') = h(j)$. We have

$$\mathbf{Pr}_h[X_j = 1] = \left(1 - \frac{1}{m}\right)^{\|x\|_0 - 1} \geq 1 - \frac{1}{\ell \cdot \ell'}.$$

Thus,

$$\mathbf{E}_h\left[\sum_{j \in \text{supp}(x)} X_j\right] \geq \left(1 - \frac{1}{\ell \cdot \ell'}\right)\|x\|_0,$$

so by Markov's inequality, we have

$$\mathbf{Pr}_h\left[\left(\|x\|_0 - \sum_{j \in \text{supp}(x)} X_j\right) \geq \frac{\|x\|_0}{\ell}\right] \leq \frac{1}{\ell'},$$

as desired. $\qquad\square$

We are now ready to prove Theorem 4.1.

*Proof of Theorem 4.1.* First we prove privacy. Consider just one fixed function $f^i_{g_j, h_j}$ for a fixed $i$ and $j$ in Algorithm 3. For simplicity, we just denote this as $f$. Let $(a_1, s_1) \ldots, (a_T, s_T)$ and $(a'_1, s'_1)$, ..., $(a'_T, s'_T)$ be two neighboring streams for either event level privacy. Then $(f(a_1), s_1) \ldots, (f(a_T), s_T)$ and $(f(a'_1), s'_1), \ldots, (f(a'_T), s'_T)$ are also neighboring streams (for the same privacy model). This is because if a single event changes in the stream from say $(a_i, s_i)$ to $(a'_i, s'_i)$, then this also induces at most a single event change in the stream after mapping under $f$ (note it could also induce no change if the domain elements $a_i$ and $a'_i$ already collide under $f$ and $s_i = s'_i$).

Thus, Algorithm 3 keeps track of $O(\log^2 T)$ different frequency estimations, each of which are $\rho'$-zCDP where $\rho'$ is an appropriate scaling of $\rho$. Then by composition via Lemma 2.1, outputting all the frequency estimates $\tilde{f}$ under all functions $f$ that we considered is $\rho$-zCDP, as desired. Note the rest of the algorithm proceeds by post-processing which is also private.

Now we argue for the approximation factor. Fix a timestep $t$. Assume for now that $\|x_t\|_0 \geq \Omega(\log^5(T)/\rho)$. The other case will be handled shortly. Now also consider one of our mappings $f$ to $[2^i]$. Lemma D.1 tells us that if $2^i \leq \|x_t\|_0/\ell$ for $\ell \geq 100\log(T)$, then for any fixed coordinate of the mapping $f$, that coordinate is larger than $\Omega(\sqrt{\ell})$ in absolute value with probability at least $97\%$. In particular, if we pick the largest $i$ such that $2^i \leq O(\|x_t\|_0/\log(T))$, then the coordinate

wise median of $f^i_{g_1,h_1}(x_t), \dots, f^i_{g_{O(\log T)},h_{O(\log T)}}(x_t)$ satisfies that all of its coordinates are larger than $\Omega(\sqrt{\ell})$ in absolute value with probability $1 - 1/\mathrm{poly}(T)$. This is still true if we consider the coordinates of $\hat{F}^i$ and set $\ell = \Theta(\log^3(T)/\rho')$, since we have additive error $\tau = O(\log^{1.5}(T)/\sqrt{\rho'})$ on every coordinate of $f^i_{g_j,h_j}(x_t)$.

Conversely, now consider the case where $2^i \geq \ell\|x_t\|_0$ for the same choice of $\ell$ as above. Then the coordinate wise median of $f^i_{g_1,h_1}(x_t), \dots, f^i_{g_{O(\log T)},h_{O(\log T)}}(x_t)$ satisfies that all of its coordinates are 0 with probability $1 - 1/\mathrm{poly}(T)$, and so $\|\hat{F}^i(t)\|_\infty < C'\tau$ by picking $C'$ sufficiently large.

To summarize, plugging in our value of $\rho'$ and a union bound, we have that any $i$ where $2^i \leq O(\|x_t\|_0\rho/\log^5(T))$ satisfies $\|\hat{F}^i(t)\|_\infty \geq C'\tau$. Conversely, any $i$ where $2^i \geq \Omega(\log^5(T)\|x_t\|_0/\rho)$ satisfies $\|\hat{F}^i(t)\|_\infty < C'\tau$. Thus, our choice of $i^*(t)$ is a multiplicative $O(\log^{10}(T)/\rho^2)$ approximation to $\|x_t\|_0$ with probability $1 - 1/\mathrm{poly}(T)$.

Now we handle the case where $\|x_t\|_0 \leq O(\log^5(T)/\rho)$. Here, the above paragraph implies that we won't return anything larger than $2^i$ where $2^i \geq \ell\|x_t\|_0$ which is $O(\log^{10}(T)/\rho^2)$, which can be subsumed into our additive error. This completes the proof. $\quad\square$

The proof of Theorem 4.2 uses similar ideas to that of Theorem 4.1.

*Proof of Theorem 4.2.* For each $m_i = 2^i$, we consider $O(\log T)$ maps $h^i_1, \dots, h^i_{O(\log T)}$ mapping $[n] \to [m_i]$ as in Lemma D.3. Each such map defines a new stream in the reduced domain $[m_i]$. Now for each map, we would like to run algorithm $\mathcal{A}$. First, we need to check that our definition of neighboring streams is still valid under a mapping. The proof of this is the same as in Theorem 4.1. To summarize, the mapping will never add any new differences between neighboring streams (but it may actually reduce the number of differences if a collision occurs).

Thus, we run $O(\log^2(T))$ many copies of $\mathcal{A}$ across all functions $h^i_j$. Naturally we must scale $\rho$ down by $O(\log^2(T))$ so that our output remain private via composition of Lemma 2.1. In parallel, we also run another private algorithm $\mathcal{B}$ with $O(\log^2(T)/\sqrt{\rho})$ multiplicative and 1 additive error (e.g. our algorithm from Theorem 3.1). Note that this also further requires scaling $\rho$ down by constant factors. We also assume $\mathcal{B}$ is correct throughout all steps $T$, which happens with probability $1 - 1/\mathrm{poly}(T)$.

Now the purported algorithm $\mathcal{A}'$ operates by post-processing the outputs of the two algorithms $\mathcal{A}$ and $\mathcal{B}$ (and thus satisfies the desired privacy guarantees). $\mathcal{A}'$ does the following: at every time step $t$, it first uses $\mathcal{B}$ to compute the smallest power of 2, denoted as $m_t$, which satisfies $O(\|x_t\|_0) \geq m_t \geq \Omega(\|x_t\|_0 \cdot \sqrt{\rho}/\log^2(T))$. Then it outputs the median of the answers released by $\mathcal{A}$, but only on the maps corresponding to the next power of 2, denoted as $m'_t$, which is larger than $O\left(\frac{m_t \log^2(T)}{\sqrt{\rho}\eta}\right)$ (so that $m'_t$ satisfies the hypothesis of Lemma D.3 as we shall see shortly).

For correctness, fix a time step $t$. Lemma D.3 implies that as long as $m'_t \geq \Omega(\|x_t\|_0/\eta)$, which is true by our choice, then the number of distinct elements under a map $h : [n] \to [m'_t]$ as in *Lemma D.3* satisfies that $\|f_h(x)\|_0$ is a $1 + \eta$ approximation to $\|x_t\|_0$ with probability 99%. In which case, the median answer across $O(\log T)$ independent copies of $h$ is thus a $1 + \eta$ approximation with probability $1 - 1/\mathrm{poly}(T)$. However, each of our calls of $\mathcal{A}$ gives additive error $\left(\frac{\|x_t\|_0}{\eta}\right)^{c_1} \cdot \mathrm{poly}(\log(T), 1/\rho)$, so our median output is a $1 + \eta$ multiplicative and $\left(\frac{\|x_t\|_0}{\eta}\right)^{c_1} \cdot \mathrm{poly}(\log(T), 1/\rho)$ additive error estimate. Since $c_1 < 1$, $\|x_t\|_0^{c_1}$ can be absorbed into the multiplicative error, if $\|x_t\|_0$ is sufficiently larger than $\mathrm{poly}(\log(T), 1/\eta, 1/\rho)$. This completes the proof. $\quad\square$

# E    PROOFS FOR SECTION 5

*Proof of Theorem 5.1.* The algorithm appears as Algorithm 4. Let $\alpha_0 > 0$ to be fixed later and let $A$ be the random matrix of the Johnson-Lindenstrauss Lemma 2.3 with $m \geq C_1(\log T)/\alpha_0^2$ with $C_1$ sufficiently large such that with high probability in $T$, it holds for all $t \in [T]$ simultaneously that if $y^{(t)} = Ax^{(t)}$, then $\left| \|y^{(t)}\|_2^2 - \|x^{(t)}\|_2^2 \right| \leq \alpha_0 \|x^{(t)}\|_2^2$ . Our algorithm uses a DP-algorithm for continual counting to estimate each entry of $y^{(t)}$. To see why this is possible, fix $i$ and denote by $A_i$ the $i$th row of $A$. Then

$$(Ax^{(t)})_i = A_i x^{(t)} = \sum_{j=1}^{n} A_{ij} x_t[j].$$

Since the entries of $A$ are $\pm 1/\sqrt{m}$, it follows that changing a single $x_t[j]$ by $\pm 1$ will change the count $(Ax^{(t)})_i$ by at most $O(1/\sqrt{m})$. To be precise, define $b_1, \ldots, b_T$ by

$$b_t = \sum_{j=1}^{n} A_{ij} x_t[j] - \sum_{j=1}^{n} A_{ij} x_{t-1}[j] = \sum_{j=1}^{n} A_{ij}(x_t[j] - x_{t-1}[j]) \in \{-1/\sqrt{m}, 0, 1/\sqrt{m}\},$$

so that $y_t^i := \sum_{s=1}^{t} b_s = \sum_{j=1}^{n} A_{ij} x_t[j]$. Then, if an adjacent dataset is obtained by replacing a single input $(a_{t_0}, s_{t_0})$ with $(a'_{t_0}, s'_{t_0})$, then defining the $b'_t$ analogously for the neighboring dataset, it is easy to check that $b_t = b'_t$ for $t \neq t_0$. For some choice of privacy parameter $\rho_0$, we may thus apply Theorem 2.1 (appropriately rescaled by $1/\sqrt{m}$), to obtain estimates $\hat{y}_t^i$ of $y_t^i$, such that $\max_{t \in [T]} |y_t^i - \hat{y}_t^i| = O(\frac{(\log T)^{1.5}}{\sqrt{\rho_0 m}})$ with high probability in $T$ and such that outputting these estimates is $\rho_0$-zCDP.

We are not quite done as we want to release estimates of $\sum_{s=1}^{t} b_t = \sum_{j=1}^{n} A_{ij} x_t[j]$ for each fixed $i \in [m]$. For this, we use the analysis above with the privacy parameter $\rho_0$ chosen such that when employing advanced composition over all $m$ releases, the final algorithm is $\rho$-zCDP. By Lemma 2.1, it follows that we can put $\rho_0 = \rho/m$ to achieve that over the $m$ releases, the final algorithm is $\rho$-zCDP. The maximum final error of a single estimate is thus $O(\frac{(\log T)^{1.5}}{\sqrt{\rho}})$. Denote by $\lambda$ this upper bound on the maximum error.

Now since the above algorithm for releasing all estimates $\hat{y}_t^i$ is $\rho$-DP, by post-processing, we may release any function of these estimates without violating privacy. Our final estimate of $\|x^{(t)}\|_2^2$ is simply $\sum_{i \in [m]} (\hat{y}_t^i)^2$. Below, we analyze the accuracy of this estimator.

**Accuracy**    Using that $(\hat{y}_t^i)^2 - (y_t^i)^2 = (\hat{y}_t^i - y_t^i)(\hat{y}_t^i + y_t^i)$, and our bound $|\hat{y}_t^i - y_t^i| \leq \lambda$, we have that

$$\left| \sum_{i \in [m]} (\hat{y}_t^i)^2 - \sum_{i \in [m]} (y_t^i)^2 \right| \leq \lambda \sum_{i \in [m]} |\hat{y}_t^i + y_t^i|$$

Let $S_1 = \{i \in [m] : |y_t^i| \leq \lambda/\alpha_0\}$ and $S_2 = [m] \setminus S_1$. Note that $|y_t^i| \leq \lambda/\alpha_0$ implies that $|y_t^i + \hat{y}_t^i| \leq \lambda(1 + 1/\alpha_0) \leq 2\lambda/\alpha_0$. Thus,

$$\sum_{i \in S_1} |\hat{y}_t^i + y_t^i| \leq \frac{2|S_1|\lambda}{\alpha_0}.$$

Moreover, $|y_t^i| > \lambda/\alpha_0$ implies that $|\hat{y}_t^i + y_t^i| \leq 2|y_t^i| + \lambda \leq 2\frac{\alpha_0}{\lambda}(y_t^i)^2 + \lambda$, and so,

$$\sum_{i \in S_2} |\hat{y}_t^i + y_t^i| \leq \frac{2\alpha_0}{\lambda} \|y_t\|_2^2 + \lambda|S_2|.$$

Combining these bounds, we get that

$$\left| \sum_{i \in [m]} (\hat{y}_t^i)^2 - \sum_{i \in [m]} (y_t^i)^2 \right| \leq 2\alpha_0 \|y_t\|_2^2 + \frac{2\lambda^2 m}{\alpha_0}.$$

It thus follows from the high probability guarantee of the JL map and the triangle inequality that,

$$\left| \sum_{i \in [m]} (\hat{y}_t^i)^2 - \|x^{(t)}\|_2^2 \right| \leq 2\alpha_0 \|y_t\|_2^2 + \frac{2\lambda^2 m}{\alpha_0} + \alpha_0 \|x^{(t)}\|_2^2 \leq (2\alpha_0(1 + \alpha_0) + \alpha_0) \|x^{(t)}\|_2^2 + \frac{2\lambda^2 m}{\alpha_0}.$$

Picking $\alpha_0 = \alpha/5$, the above bound is at most $\alpha \|x^{(t)}\|_2^2 + \frac{10\lambda^2 m}{\alpha}$. Plugging in the values of $m$ and $\lambda$ gives the desired result.

**Space Usage**   It is well known that the dimensionality reduction matrix $A$ satisfies the guarantee of Lemma 2.3 even if the entries are only $O(\log T)$ independent Clarkson & Woodruff (2009). To populate the entries of $A$, we can for example use $k$-independent hashing Wegman & Carter (1981) which when implemented as polynomials of degree $k = O(\log T)$, requires $O(\log T)$ words of memory. We additionally, need to store the $m$ counters for continual counting. By Theorem 2.1, this can be achieved with $O(m \log T) = O((\log T)^2/\alpha^2)$ words of memory.   □

