# OpenReview forum: "Skirting Additive Error Barriers for Private Turnstile Streams"
_ICLR.cc/2026/Conference — ICLR 2026 Poster_

### Official Review · Reviewer_RNQM · 2025-10-23

**Soundness:** 3
**Presentation:** 3
**Contribution:** 3
**Rating:** 8
**Confidence:** 3

**Summary:**

The paper studies (approximately) differentially private (DP) release of the number of distinct items in a turnstile stream, where at each step an item can be either added or removed from the active set. The stream has length $T$, the universe of items is of size $n$, and the notion of neighboring is *event-level*, meaning that the input at a single step can change. Past work showed a lower bound of $\Omega(T^{1/4})$ (Jain et al., 2023a) (event-level) and gave an algorithm running in $\tilde{O}(T^{1/3})$ (Jain et al., 2023b) (item-level), both on the additive error achieved by the algorithm. This paper circumvents the lower bound by considering mixed error guarantees which are also *multiplicative*, showing that $\mathrm{polylog}(T)$ additive error is possible if $\mathrm{polylog}(T)$ multiplicative error is tolerated. This can be contrasted with other lines of work (Jain et al., 2023a; Henzinger et al., 2024a; Cummings et al., 2025) that instead pursue guarantees parameterized in the input.

The paper also investigates the $F_2$ moment on turnstile streams, showing that any purely additive error guarantee necessarily is $\Omega(T)$, but that if constant multiplicative error is allowed then $\mathrm{polylog}(T)$ additive error is possible.

The techniques in the paper are based on sketching, where the counting problem on the turnstile stream is reduced to running many parallel instances of differentially private continual counting. Algorithm 2 builds on MinHash, Algorithm 3 (conceptually in my view, see later question) on CountSketch and Algorithm 4 on the AMS sketch for $F_2$ estimation.

**Strengths:**

1. The problem of (DP) counting distinct elements on a turnstile stream is easy to motivate, and a fundamental problem in DP. Given the existence of a polynomial lower bound on the additive error, studying the problem under a multiplicative guarantee is natural and well-motivated.
2. The paper is overall well-written, with the main part of the paper giving enough intuition for the results while still giving broader context.
3. The techniques/ideas used are natural, and the proofs in the appendix that I checked seemed correct (up to constants, see questions).
4. The message that polynomial additive errors can be replaced by polylogarithmic additive errors at the expense of (potentially non-trivial) multiplicative error, and sometimes in low space, is relatively clean and nice. I think this message, in combination with the Open Problems section, will lead to additional future work in this direction.
5. The subject fits the scope of ICLR.

**Weaknesses:**

1. A $\mathrm{polylog}(T)$ multiplicative guarantee will not always be competitive with the pure additive error guarantees parametrized in e.g., flippancy (Jain et al. (2023a)) or total flippancy (Henzinger et al. (2024a)). Many natural streams are prone to exhibit low flippancy (e.g., tracking occupancy in a store). The multiplicative guarantee can potentially offer an improvement in pathological cases where the flippancy is very high, e.g., when updates are concentrated in a few items, but it is not clear to me how realistic/interesting a setting this is.
2. The paper only considers event-level DP, rather than item-level DP. It is not clear how (or if) the techniques could be extended to work for item-level DP.
3. The techniques used in the paper, and how they compare to techniques used in past work on DP counting distinct elements on turnstile streams, are insufficiently discussed. This paper is not the first to use sketching in the context of DP more broadly, see e.g., [1,2,3,4]. [1] for example appears to solve the same problem in the setting where only the final number of distinct elements has to be released (i.e., not under continual release), and does so with a multiplicative/additive guarantee. More related still, the key building block in Epasto et al. (2023) is an implementation of CountSketch under continual release. It would strengthen the paper to discuss the extent to which its techniques differ from past work. This could mostly still be done in the appendix by extending Section A.

References:
- [1] The Flajolet-Martin Sketch Itself Preserves Differential Privacy: Private Counting with Minimal Space, Smith et al., NeurIPS‘20.
- [2] Differentially Private Linear Sketches: Efficient Implementations and Applications, Zhao et al., NeurIPS’22.
- [3] Improved Utility Analysis of Private CountSketch, Pagh & Thorup, NeurIPS’22.
- [4] Better Differentially Private Approximate Histograms and Heavy Hitters using the Misra-Gries Sketch, Lebeda & Tetek, ACM Trans. Datab. Syst.’25.

**Questions:**

Overall, I find the paper well-written and interesting with a meaningful contribution. I also think it invites clear follow-up work. I recommend it for acceptance, but I have some concerns regarding the novelty of the techniques used (hence my confidence score). I think the paper would benefit from, at least a brief, discussion of how the techniques employed compare to past work.

1. How novel are your techniques over past work for counting distinct elements on turnstile streams specifically? How do your techniques differ?
2. Can Algorithm 3 be viewed as continual release of CountSketch, in a “high-collision regime”? If so, how does it compare to the continual release of CountSketch employed in Epasto et al. (2023)?
3. Do you believe your techniques could be extended to work for item-level DP?
4. Not a severe issue, but I think the inequality you state on line 856 in the proof of Lemma C.1 is wrong. My understanding of the argument is that you enlarge the unit-sized interval to have size $\sqrt{l}/500$, and then union bound over ~this number of unit-sized intervals, but then the first factor on the right-hand-side should be divided by 500 rather than 50000, right? If this is an error, I think it should only impact constants in Lemma C.1, nothing in the main paper.

*Typos/comments:*
- General comment: The exact assumption on the hash/random functions involved are not stated clearly. E.g., in Algorithm 1, Step 2 it is a “random hash function” from $[n]$ to $[n]$, Lemma C.1 only specifies that $f$ and $g$ are “random functions”. From the proof of the space usage in Theorem 3.1, you state that pairwise independent hash functions are enough for the analysis in that case, and necessary for the claimed space usage, but it is not stated in the main theorem. It would benefit the clarity of the paper if the requirement on the hash functions was more explicit. Perhaps it could be stated once that e.g., "all hash functions are assumed pairwise independent and can be stored in space...", if the requirement is consistent throughout the paper.
- Line 043-044: References appear twice?
- Line 150-151: remove “to”
- Line 318-319: The interpretation of $f_t[k]$ seems wrong to me, that it is the number of elements at time $t$ whose value hash into $[2^k, 2^{k+1})$. I think this would only be true if it tracked the *most* significant bit.
- Line 393:  Step 3 of Algorithm 3 missing a “for”
- Line 444: If it *is* possible
- Line 485: Remove “for” in “algorithm for behind”. Stylistic preference “that it satisfies” -> “it satisfying”.
- Line 820: *form* the foundation..
- Line 843: [k]now
- Line 851 and 855: I think it would not hurt to add a bit of motivation for why these inequalities hold. The first seems like a loose invocation of a bound by Erdos and the second a union bound.
- Line 847: Again, it is not just any random function.
- Line 872: Large[r]

---

> ### Author Response · Authors · 2025-11-19
>
> Thanks very much for your thoughtful review! We address several specific points below.
>
> > The multiplicative guarantee can potentially offer an improvement in pathological cases where the flippancy is very high, e.g., when updates are concentrated in a few items, but it is not clear to me how realistic/interesting a setting this is.
>
> Our guarantee could be preferred over the purely additive error guarantee in two natural cases.
>
> The first case is where the number of distinct elements is less than $T^{⅓}$ in some time steps. For such time steps, we would output a non-trivial estimate, whereas the $O(T^{⅓})$ additive error guarantee algorithm would output an estimate that has a polynomial multiplicative error. We believe that streams where the true frequency vector is sparse at some intermediate timesteps are quite natural. Even if this is not the case over the entire stream, our algorithm could be run alongside a purely additive error algorithm to achieve the best of the two results (if our algorithm estimated the true value to be above $\tilde{O}(T^{1/3})$ at any time, we could switch to a purely additive error algorithm).
>
> The second case is when the length of the stream, $T$ is polynomially larger than $n$, the universe size. This is a standard regime for traditional streaming algorithms. If $T > n^{3}$, then the prior additive error is always worse than always outputting $n$ or $0$, which is a trivial estimate since the universe size is clearly an upper bound on the total number of distinct elements. On the other hand, our bound gives a non-trivial estimate all the way up to the regime where $T$ is super-polynomial in $n$.
>
> > The techniques used in the paper, and how they compare to techniques used in past work on DP counting distinct elements on turnstile streams, are insufficiently discussed. This paper is not the first to use sketching in the context of DP more broadly, see e.g., [1,2,3,4].
>
> Thanks for pointing out these related works. We will include a more detailed discussion in Appendix A.
>
> The setting in [1] only releases a final number and does not handle our continual release setting. The version where we only release a final number is quite easy for DP (ignoring space constraints) since the sensitivity of the final value is only 1 and should not require any sketching techniques. [2,3] don’t consider the distinct element problem but rather the frequency estimation problem.  While not considering the continual release setting, they show that simple modifications of the classic algorithms CountMin and CountSketch suffice to ensure privacy and small error. Classic CountSketches can be used for estimating $F_2$-moments (in fact, the JL transform we use basically consists of $m$ count-sketches). However, while [2,3], allow us to release *all* frequencies with no additional privacy loss, their work only provides a uniform guarantee on the error of each of the individual frequency estimates. It seems unlikely that such a guarantee, can be used to get $F_2$-estimates of a similar guarantee to ours.
>
> [4] considers a private version of the Misra-Gries sketch which is used to solve the heavy-hitters problem (detecting large frequencies). Heavy-hitters and distinct elements are measuring two different statistics of the underlying frequency vector and to the best of our knowledge, estimating heavy hitters does not imply any meaningful estimate of the number of distinct elements. Misra-Gries is also a deterministic sketch and does not use hashing techniques used in our paper.
>
> The main technical challenge in our work is handling the continual release setting which requires us to reduce to a setting where we can use DP continual counting algorithms (e.g. by bucketing using a hash function). We are not aware of related work that uses similar techniques together with continual counting.
>
> > How novel are your techniques over past work for counting distinct elements on turnstile streams specifically? How do your techniques differ?
>
> We agree that our techniques make use of some of the most basic and fundamental toolkits in both streaming and differential privacy. However, as is often the case in DP, the choice of how to privatize an appropriate algorithm is often unclear. For example, consider the standard textbook minimum hash estimator for distinct elements. It hashes stream elements uniformly at random into [0,1] and uses 1/(minimum hashed value) as an estimator for distinct elements. A naive calculation would upper bound the sensitivity of the minimum hashed value by O(1) (since the value is always in the interval [0, 1]), but this is not strong enough to be useful in the continual setting as the minimum hashed value could change often over the course of the stream as the result of a single event-level change.

---

> > ### Author Response · Authors · 2025-11-19
> >
> > > Can Algorithm 3 be viewed as continual release of CountSketch, in a “high-collision regime”? If so, how does it compare to the continual release of CountSketch employed in Epasto et al. (2023)?
> >
> > The answer is both yes and no: Epasto et al. (which study the insertion only streams) also use the fact that frequencies can be estimated using continual counting. However, their main CountSketch guarantee (see Theorem 1.4 in https://arxiv.org/pdf/2301.05605) is not directly useful for distinct elements. This is because the theorem guarantees that every frequency is estimated up to error that depends on the $\ell_2$ norm of the true underlying frequency vector at that time step (plus some poly(logarithmic) additive error). The additive $\ell_2$ error can be polynomial in $T$ in the worst case and is too coarse to detect 0 versus non-zero frequencies.
> >
> > However, the underlying mechanism in CountSketch is the same simple hashing as in our Algorithm 3. Traditional CountSketch analysis tries to upper bound collisions, as collisions are ‘bad’ events which distort frequency values. However, in our case, we desire the opposite behavior. We need to ensure sufficiently many collisions occur to increase the frequencies (after hashing) to be above the continual counting noise floor. Since Algorithm 3 handles general turnstile streams (where entries can be negative), we also need to use anti-concentration inequalities to guarantee that the hashed frequencies don’t sum to be close to 0. Note that this property can also be obtained by hashing to a domain of size one which would not be useful in our setting. Thus, we additionally show that if the reduced dimension is sufficiently *large* compared to the true number of distinct elements, then we can avoid having many collisions, thereby preserving the original value of the true number of distinct elements. Finally the formal proof of the guarantees of Algorithm 3 proceeds by combining these two results into a “middle ground,” where sufficiently many non zero frequencies collide which allows for estimation, while still ensuring that the original number of distinct elements has not been reduced by more than poly-logarithmic factors.
> >
> > > Do you believe your techniques could be extended to work for item-level DP?
> >
> > This is a good question. Our techniques fail in this setting as they use frequency counts via DP continual counters as intermediate objects which have very large sensitivity if all item occurrences in a stream can change, so the item-level DP setting likely requires new ideas.
> >
> > > Not a severe issue, but I think the inequality you state on line 856 in the proof of Lemma C.1 is wrong. My understanding of the argument is that you enlarge the unit-sized interval to have size, and then union bound over ~this number of unit-sized intervals, but then the first factor on the right-hand-side should be divided by 500 rather than 50000, right? If this is an error, I think it should only impact constants in Lemma C.1, nothing in the main paper.
> >
> > Thanks for pointing this out, we will update the paper. This does indeed not impact our results by more than constant factors.
> >
> > > General comment: The exact assumption on the hash/random functions involved are not stated clearly. E.g., in Algorithm 1, Step 2 it is a “random hash function” from to , Lemma C.1 only specifies that and seems wrong to me, that it is the number of elements at time whose value hash into
> >
> > We agree that the paper would benefit from a more explicit statement and discussion of the properties we require of the hash functions and will update the paper accordingly. It is indeed sufficient for theorem 3.1 that the hash functions are pairwise independent. It is not the case for all our results that pairwise independent hashing suffices. For instance, for Theorem 5.1, we populate the random matrix $A$ with using $O(\log T)$ independent hashing. See the discussion of l1049-l1055.
> >
> > > Misc. typos
> >
> > Thanks very much for making us aware of the remaining typos. We will fix them in the next version of the paper.

---

> > > ### Comment · Reviewer_RNQM · 2025-11-24
> > >
> > > Thank you for the clarifications and answering my questions. I stand by my score.

---

### Official Review · Reviewer_Azzh · 2025-11-03

**Soundness:** 3
**Presentation:** 3
**Contribution:** 2
**Rating:** 6
**Confidence:** 4

**Summary:**

This paper considers the differentially private estimation of the fundamental stream quantity of distinct elements (and F2 norm). The privacy (and utility) guarantee is with respect to all intermediate updates of the number of distinct elements. This is kknown as the continual release model of differential privacy.

Prior work shows an $T^{1/3}$-additive error algorithm, where $T$ is the stream length, for continually counting the number of disinct elements in turnstile streams, where stream updates consist of elements insertions and removals. They also show a lower bound of $T^{1/4}$ on the additive error. (The problem is much simpler for insertion-only streams, where polylog T error is achievable). The authors of this paper show that by allowing mulitplicative  error, we can bypass the polynomial lower bound on the additive error. They show an algorithm with polylogT additive and mulitplicative error that also has polylogT space (the prior work used polynomial in T space).

An open question on the optimality of the multiplicative error remains, in particular can there be algorithms with constant multiplicative error for this private estimation problem?

**Strengths:**

- This paper opens an interesting research direction of considering multiplicative error algorithms in the space of continual release algorithms, where strong additive error lower bounds are known.

- Allowing for multiplicative error gives rise to algorithms with polylog space, which has been a relevant question of prior works (Jain et al 2023, Cummings et al 25).

**Weaknesses:**

- The algorithmic techniques are not very novel, the main results are achieved through a combination of common streaming algorithms and the differentially private technique of continual counting.

- Without lower bounds, it is hard to say how close/far from optimal the approximation bound is. It is very open whether an algorithm with constant multiplicative error exists.

**Questions:**

The algorithm in Section 4 was hard to grasp without much explanation in words. Could you please provide an overview of the technical novelty for Algorithm 4 compared to existing work in the non-private literature?

---

> ### Author Response · Authors · 2025-11-19
>
> Thanks very much for your thoughtful review! We address the specific points below.
>
> > The algorithmic techniques are not very novel, the main results are achieved through a combination of common streaming algorithms and the differentially private technique of continual counting.
>
> We agree that our techniques make use of some of the most basic and fundamental toolkits in both streaming and differential privacy. However, as is often the case in DP, the choice of *how* to privatize an appropriate algorithm is often unclear. For example, consider the standard textbook minimum hash estimator for distinct elements. It hashes stream elements uniformly at random into [0,1] and uses 1/(minimum hashed value) as an estimator for distinct elements. A naive calculation would upper bound the sensitivity of the minimum hashed value by O(1) (since the value is always in the interval [0, 1]), but this is not strong enough to be useful in the continual setting as the minimum hashed value could change often over the course of the stream as the result of a single event-level change.
>
> > Without lower bounds, it is hard to say how close/far from optimal the approximation bound is. It is very open whether an algorithm with constant multiplicative error exists.
>
> We agree that for distinct element estimation, it is indeed an intriguing open question to determine if one can improve our approximation factor to a constant. However we note that our other main result about F2 estimation already achieves a $1+\eta$ multiplicative approximation in polylogarithmic space, which is the best one can hope for, even in the non private setting.
>
> > The algorithm in Section 4 was hard to grasp without much explanation in words. Could you please provide an overview of the technical novelty for Algorithm 4 compared to existing work in the non-private literature?
>
> At a high-level, the algorithm of Section 4 is giving a dimensionality reduction map which preserves the number of distinct elements up to poly-logarithmic multiplicative error. Unlike traditional dimensionality reduction however, we are not necessarily interested in reducing the dimension of the frequency vector (which is the size of the universe). Rather, we exploit a different property of dimensionality reduction which is that it increases the value of the coordinates in the reduced dimension. This is useful from a DP perspective, as large coordinates can be detected via continual counting.
>
> More specifically, we prove three structural lemmas. Lemma C.1 says that if the reduced dimension is sufficiently *small* (compared to the true number of distinct elements), then all coordinates in the reduced dimension become large. Lemma C.2 and C.3 together imply a converse result: if the reduced dimension is sufficiently *large* compared to the true number of distinct elements, then we can avoid having many collisions, thereby preserving the original value of the true number of distinct elements. Finally the formal proof of Algorithm 3 proceeds by combining these two results into a “middle ground,” where sufficiently many non zero frequencies collide which allows for estimation, while still ensuring that the original number of distinct elements has not been reduced by more than poly-logarithmic factors. Note that one advantage of this algorithm is that it also works in the setting where frequencies are allowed to be negative.

---

> > ### Comment · Reviewer_Azzh · 2025-11-23
> >
> > Thank you for the clarification and apologies for my oversight on the F2 estimator with constant multiplicative error, my comment was focused on counting distinct elements. I will retain my score and encourage the authors to use the additonal page in the final paper draft to provide more intution about Algorithm 4 -- the intuition provided in your comment was very helpful.

---

### Official Review · Reviewer_NCxW · 2025-11-03

**Soundness:** 3
**Presentation:** 3
**Contribution:** 3
**Rating:** 6
**Confidence:** 3

**Summary:**

The paper studies the problem of estimating statistics on a data stream under a general turnstile model. In this model, there is a strong lower bound that essentially rules out any algorithm with additive error that is logarithmic in the stream length. The problem is also studied in other models of streams, specifically insertion-only models, where one can achieve a polylog additive error with a small multiplicative approximation.

The paper gives algorithms for distinct element counting and F_2 moment estimation with additive error that scales with poly-log(T) at a cost of a large multiplicative error.

**Strengths:**

The paper's strength lies in being the first private algorithm for distinct counting and F_2 moment estimation under the general turnstile update model. Their result also raises a general question as to what the tradeoff is between multiplicative and additive error. I also like the fact that the authors mention some open problems with proper discussion.

In terms of idea, I do not see something new that comes up and that is alright. I do not consider it a weakness and actually consider a strength with a small bias (so mentioning it in the strengths section).

**Weaknesses:**

My biggest worry is that the multiplicative error and the additive error have the same scale. In particular, what the authors end up showing in the upper bound side is that they can approximate distinct elements with polylog(T) multiplicative approximation (the additive term can get subsumed if we consider that there are at least $O(1)$ distinct elements). The same goes for $F_2$ estimation. This makes me a little worried about the strength of the result, especially because in both problems, one can achieve a significantly smaller multiplicative approximation (without privacy constraints). Since the extra $\log^2(T)$ factor in space comes mainly because of continual release, we can take $\rho = O(\log^4(T))$ and get a constant approximation, but larger space than non-private variants. Ideally, one sanity check I like to make is to let the privacy parameter go to $\infty$, and we should recover the result in a non-private setting. That does not seem to happen here. Can the authors shed some light here?

**Questions:**

Please look at my weakness point above.

---

> ### Author Response · Authors · 2025-11-19
>
> Thanks very much for your thoughtful review! We address the one weakness/question below.
>
> We agree that it is indeed a natural goal to obtain private algorithms that obtain the same guarantees as their classical counterparts as the privacy parameters turn to infinity. However, to the best of our knowledge, even in the simple continual counting setup it is not known how to obtain such guarantees. There, the binary tree mechanism uses a factor of $\log(T)$ more space than what is needed in the non-private setting even as the privacy parameter goes to infinity ($\log(T)$ counters must be maintained while non-privately, only one counter is needed). This is therefore a tough criterion and, while interesting, beyond the scope of our approaches which utilize DP continual counters as subroutines.

---

### Official Review · Reviewer_3SKD · 2025-11-09

**Soundness:** 4
**Presentation:** 3
**Contribution:** 3
**Rating:** 6
**Confidence:** 3

**Summary:**

The paper provides new algorithms for differentially-private, low-space estimators of statistics on a **turnstile** data stream, where elements may be added and removed in the stream. The paper gives algorithms with a mixed multiplicative $polylog(T)$ and additive $poly(log(T)/\epsilon)$ guarantee for counting distinct elements and $F_2$ estimation. These algorithm build on a pair of recent results, one that uses linear space to get an additive-only guarantee, and another that uses sublinear but still polynomial in T (about $T^{1/3}$) space.

On a technical level, the paper modifies existing sketching algorithms. They obtain significantly stronger results in the `strict turnstile' setting (where deletion requests only occur for items that actually exist in the current logical dataset) than in the general setting.

**Strengths:**

* The paper provides a nice contribution to the literature on space-bounded private computations, addressing basic problems ($F_0$ and $F_2$ estimation).

* The paper an open question explicitly asked by previous work of JKRSS (Jain et al, ICML 2023); it improves on [CEMMOZ] (Cummings et al., ICML 25), which explicitly addressed the same question.

* The paper appears to be technically sound, with clear high-level overviews of the algorithms and proof ideas.

**Weaknesses:**

* The polylog multiplicative guarantee is quite weak. The previous work of CEMMOZ (as well as a combination of JKRSS and KNW'10) achieves a $(1+\eta)$-multiplicative guarantee, and already demonstrates that sublinear additive error and space guarantees are possible. Is there any evidence that one cannot get a constant-factor multiplicative approximation along with a polylogarithmic additive error guarantee?


Significant, but not major, drawbacks:

* The techniques in the paper are standard—they mostly consist of combining known tools in the right way. (There's nothing wrong with that, but it means that new technical tools are not a big contribution here.)

* The fit for ICLR is a bit weird. Although cardinality estimation is a basic algorithmic and statistical topic, it's unclear how relevant it is to most of the ICLR audience. Of course, ICLR is quite broad at this point.

* The comparison to previous work in Table 1 is missing a discussion of the work of [CEMMOZ]. That work provides the first sublinear-space algorithm for the problem (with a much tighter multiplicative guarantee and a much looser additive error and space guarantee than the ones in this paper). If I understood correctly, this submission points out that one could achieve a better result (that is, $(1+\eta, \tilde O(\sqrt[3]{T}))$ multiplicative/additive error in polylog space) by directly combining the JKRSS approach with an algorithm due to Kane et al. It's good to point that out, but I would still keep CEMMOZ in the discussion. (Also, it would be good to spell out the combination of JKRSS with Kane et al in a bit more detail—perhaps in the appendix—and clarify the attribution in Table 1, since the result does not appear in JKRSS.)


Minor comments:

* The exact (event-level) privacy definition is unclear until page 3 of the paper. Claiming "differential privacy" doesn't make sense without an adjacency notion. Given that several have been studied for this problem, it makes sense to clarify the point early.

* The theorem statements claim 1/polylog(T) probability of error. Presumably one could amplify this by running several parallel copies of the algorithm (and increasing epsilon). Why the weaker statement?

**Questions:**

* What is possible with item-level guarantees? (These are natural for cardinality, which is 1-sensitive to item-level changes at every time step.)

---

> ### Author Response · Authors · 2025-11-19
>
> Thanks very much for your thoughtful review! We address several specific points below.
>
> > Is there any evidence that one cannot get a constant-factor multiplicative approximation along with a polylogarithmic additive error guarantee?
>
> This is an intriguing open question which we hope will be a fruitful direction for future work. The core of our approaches is to use DP continual counting within schemes that (classically) are designed to give multiplicative approximations. This idea runs into the barrier that the counters only provide estimates within an additive $\text{polylog}(T)$ which in turn blows up the multiplicative error beyond a constant factor. For instance, for our first algorithm, this essentially means that we cannot distinguish between a hash bucket containing (a) one element appearing $\text{polylog}(T)$ times or (b) $\text{polylog}(T)$ elements each occurring once. We are hopeful that the constant-factor multiplicative approximation along with a polylogarithmic additive error guarantee should be achievable, but would likely require new ideas.
>
> > The fit for ICLR is a bit weird.
>
> We would like to point out the following papers which study private continual estimation problems which appeared in the top ML conferences in the last two years:
>
> Joel Daniel Andersson and Rasmus Pagh. A smooth binary mechanism for efficient private continual observation. NeurIPS 23
>
> Joel Daniel Andersson, Monika Henzinger, Rasmus Pagh, Teresa Anna Steiner, and Jalaj Upadhyay. Continual counting with gradual privacy expiration. NeurIPS 24
>
> Rachel Cummings, Alessandro Epasto, Jieming Mao, Tamalika Mukherjee, Tingting Ou, and Peilin Zhong. Differentially private space-efficient algorithms for counting distinct elements in the turnstile model. ICML 25
>
> Hendrik Fichtenberger, Monika Henzinger, and Jalaj Upadhyay. Constant matters: fine-grained error bound on differentially private continual observation. ICML 23
>
> Palak Jain, Iden Kalemaj, Sofya Raskhodnikova, Satchit Sivakumar, and Adam Smith. Counting distinct elements in the turnstile model with differential privacy under continual observation. NeurIPS 2023
>
> Palak Jain, Sofya Raskhodnikova, Satchit Sivakumar, and Adam Smith. The price of differential privacy under continual observation. NeurIPS 2023
>
> > The comparison to previous work in Table 1 is missing a discussion of the work of [CEMMOZ]...
>
> Thanks so much for this comment. We will highlight the work of CEMMOZ in the table and discussion. In fact, it is not clear that one can combine the JKRSS approach with the algorithm by Kane et al. The statement that this is possible is due to an oversight on our part. The issue is that the $(1+\eta)$ multiplicative factor algorithm by Kane et al. does not have low sensitivity, so even releasing its output at a single point of time would a priori require adding noise scaling with $\eta F_0$. We are very sorry for this oversight, and will update the paper accordingly.
>
> > The exact (event-level) privacy definition is unclear until page 3 of the paper...
>
> We agree and will discuss the adjacency setting earlier in the paper.
>
> > The theorem statements claim 1/polylog(T) probability of error...
>
> Thanks for pointing this typo out. It should indeed have read failure probability $1/poly(T)$ as is correctly stated in the full versions of the theorems.
>
> > What is possible with item-level guarantees?
>
> This is a good question for future work. Our techniques fail in this setting as they use frequency counts via DP continual counters as intermediate objects which have very large sensitivity if all item occurrences in a stream can change.

---

### Meta-Review · Program_Chairs · 2026-01-07

**Summary:**

This paper is accepted for providing the first sublinear-space algorithms for $F_0$ and $F_2$ estimation in the strict turnstile model under continual observation. The work successfully addresses open questions from recent literature, demonstrating that polylogarithmic additive error is achievable by strategically adapting sketching techniques for privacy.

Points of Support

**Reviewer Concerns:**

Resolution of Open Problems: Directly answers challenges posed by JKRSS (2023) and improves upon recent benchmarks like CEMMOZ.

Technical Soundness: Reviewers confirmed the proofs are rigorous and the high-level algorithmic overviews are clear.

Theoretical Impact: Establishes a new baseline for the tradeoff between multiplicative and additive error in private streaming.

Constructive Rebuttal: Authors clarified technical oversights regarding item-level sensitivity and corrected erroneous attributions in the comparison table.

**Reviewer Scores:**

NA

---

### Decision · Program_Chairs · 2026-01-26

Accept (Poster)